# Photonic Stochastic Emergent Storage for deep classification by scattering-intrinsic patterns

**Marco Leonetti** [1,2,3] ✉, **Giorgio Gosti** [1,2,4] **& Giancarlo Ruocco**[2,5]

Disorder is a pervasive characteristic of natural systems, offering a wealth of non-repeating patterns. In this study, we present a novel storage method that harnesses naturally-occurring random structures to store an arbitrary pattern in a memory device. This method, the Stochastic Emergent Storage (SES), builds upon the concept of emergent archetypes, where a training set of imperfect examples (prototypes) is employed to instantiate an archetype in a Hopfield-like network through emergent processes. We demonstrate this non-Hebbian paradigm in the photonic domain by utilizing random transmission matrices, which govern light scattering in a white-paint turbid medium, as prototypes. Through the implementation of programmable hardware, we successfully realize and experimentally validate the capability to store an arbitrary archetype and perform classification at the speed of light. Leveraging the vast number of modes excited by mesoscopic diffusion, our approach enables the simultaneous storage of thousands of memories without requiring any additional fabrication efforts. Similar to a content addressable memory, all stored memories can be collectively assessed against a given pattern to identify the matching element. Furthermore, by organizing memories spatially into distinct classes, they become features within a higher-level categorical (deeper) optical classification layer.

Neural networks have made significant contributions to the field of Artificial Intelligence, serving as both a tool for mathematical modeling and a means to understand brain function. The Hopfield paradigm[1,2] has played a crucial role in this domain, utilizing a synaptic matrix to represent the interconnections between neurons. This matrix possesses the remarkable ability to store and recognize patterns, and serve as a fundamental framework for the realization of future content-addressable memory (CAM)[3,4].

To store a memory consisting of $N$ elements, the widely adopted approach is to employ Hebb's rule[5]. This rule entails constructing a synaptic matrix, denoted as $\boldsymbol{T}$, by taking the tensorial product of the vector $\boldsymbol{\phi}^*$ (representing the pattern to be stored) and its conjugate transpose ($\boldsymbol{\phi}^{*\dagger}$):

$$\boldsymbol{T} = \boldsymbol{\phi}^* \otimes \boldsymbol{\phi}^{*\dagger} \qquad (1)$$

However, there is a fundamental limit to the number of memories that can be reliably stored using Hebbian-based approaches. As the network becomes more densely populated, the interactions between different memory elements can lead to the emergence of unintended and uncontrolled memory states[2]. To address this limitation, recent research has explored various methods to enhance the capacity of neural networks: dilution[6–9], autapses[10,11], and convex probability flow[12,13].

[1]Soft and Living Matter Laboratory, Institute of Nanotechnology, 00185 Rome, Italy. [2]Center for Life Nano- & Neuro-Science, Italian Institute of Technology, Rome, Italy. [3]Rebel Dynamics-IIT CLN2S Jointlab, 00161 Roma, Italy. [4]Istituto di Scienze del Patrimonio Culturale, Sede di Roma, Consiglio Nazionale delle Ricerche, 00010 Montelibretti (RM), Italy. [5]Department of Physics, University Sapienza, I-00185 Roma, Italy. ✉e-mail: marco.leonetti@cnr.it

Recently, it was proposed to leverage the interaction among stored patterns in a constructive way: an emergent archetype may be stored by proposing to the network multiple prototypes that closely resemble the target pattern but are intentionally corrupted or filled with errors. The interaction between these prototypes serves to strengthen the emergence of the desired memory[14]. This paradigm is connected to the prototype concept developed in hierarchical clustering, in which prototypes are elements of the dataset representative of each cluster[15].

In this study, we introduce a novel learning strategy called *Stochastic Emergent Storage* (SES). SES taps into the abundance of natural randomness to construct an emergent representation of the desired memory. Capitalizing a vast database of fully random patterns freely produced by a disordered, self-assembled structure, we select a set of prototypes that bear resemblance to the target memory through a similarity-based criterion. Subsequently, by performing a weighted sum of the synaptic matrices corresponding to these selected prototypes, we are able to effectively generate the desired pattern in an emergent fashion.

Given the inherent advantages of photonic computation, such as ultra-fast wavefront transformation and parallel operation, it results that optics is the ideal domain to explore the SES paradigm. The convergence of photonics, artificial intelligence, and machine learning represents a highly active and promising area of research[3,16], leading to novel interdisciplinary paradigms such as Diffractive Deep neural networks[17,18] photonic Ising machines[19] and photonic Boltzmann computing Machines[20]. However, these approaches typically rely on direct control over optical properties of millions of scattering elements, which can be challenging and costly both with microfabrication or adaptive optical elements.

In a departure from traditional approaches, disordered scattering structures have been proposed as a radically different avenue for optical computation in various applications: classification[21], vector-matrix multiplication[22], computation of statistical mechanics ensembles dynamics[23], and others[24].

Here, we propose to employ the scattering intrinsic patterns, the optical transmission matrices, to realize a SES-based optical hardware, the disordered classifier. This device is capable of efficiently performing pattern storage, and subsequent pattern retrieval. It is able to simultaneously compare an input pattern with thousands of stored elements, and it enables a two-layer architecture, providing categorical (deep) classification, which allows for more complex tasks.

## Results

The idea stems from the fact that intensity scattered by a disordered medium into a mode $\nu$ resulting from an input pattern $\boldsymbol{\phi}$) may be written as:

$$I^\nu(\boldsymbol{\phi}) = \boldsymbol{\phi} \cdot \mathbf{V}^\nu \cdot \boldsymbol{\phi}^\dagger \qquad (2)$$

with the scattering process driven by the matrix $\mathbf{V}^\nu \in \mathbb{C}^{N \times N}$:

$$\mathbf{V}^\nu \sim \boldsymbol{\xi}^\nu \otimes \boldsymbol{\xi}^{\nu\dagger} \qquad (3)$$

generated from the tensorial product of the transmission matrix row (transmission vector) $\boldsymbol{\xi}^\nu$ ($\in \mathbb{C}^N$) with its conjugate transpose $\boldsymbol{\xi}^\dagger$.

Indeed $I^\nu(\boldsymbol{\phi})$ is maximized if $\boldsymbol{\phi} \| \boldsymbol{\xi}^\nu$: this paradigm is at the basis of the wavefront shaping techniques[25,26], in which the input pattern is adapted to the transmission matrix elements. Thus, scattering into a mode (corresponding to one of our camera pixels, see methods) is described by the same mathematics of the Hopfield Hamiltonian and a pattern is "recognized" (produces maximal intensity) if it matches the $\boldsymbol{\xi}^\nu$ vector. Given this mapping, $\mathbf{V}^\nu$ may be named an optical synaptic matrix relative to the $\boldsymbol{\xi}^\nu$ memory.

In naturally occurring scattering, one has no control over the pattern $\boldsymbol{\xi}^\nu$ and the relative optical synaptic matrix $\mathbf{V}^\nu$ because it results from a multitude of subsequent scattering events with micro-nano particles of unknown shape, optical properties, and location. Here, we propose to store an arbitrary, user-defined, memory (or pattern) in naturally occurring scattering media, by exploiting the fact that a scattering process generated billions of output modes, each with a unique and random embedded memory pattern $\boldsymbol{\xi}^\nu$ and the relative $\mathbf{V}^\nu$. Thus we propose a new method to realize a photonic linear combination of $\mathbf{V}^\nu$ to generate an artificial, (user-designed) optical synaptic matrix. This method is based on the realization of a sensor collecting the *transformed intensity*

$$I^{\mathcal{M}}(\boldsymbol{\phi}) = \sum_\nu^M \lambda^\nu I^\nu(\boldsymbol{\phi}) \qquad (4)$$

resulting from the incoherent sum of *M* intensities realized from that many transmitted optical modes from $\mathcal{M}$ which is a subset of all the modes monitored $\mathcal{M}^L$. Coefficient $\lambda^\nu$ ($\in \{0 - 1\}$ and identified by a 4 bit positive real number) represent attenuation coefficients realized by mode-specific neutral density filters. Then employing the Eq. (2) in Eq. (4) we obtain

$$I^{\mathcal{M},\lambda}(\boldsymbol{\phi}) = \boldsymbol{\phi} \cdot \left( \sum_\nu^M \lambda^\nu \mathbf{V}^\nu \right) \cdot \boldsymbol{\phi}^\dagger = \boldsymbol{\phi} \cdot \mathbf{J}^{\mathcal{M},\lambda} \cdot \boldsymbol{\phi}^\dagger. \qquad (5)$$

Then, we propose two techniques to design the optical operator $\mathbf{J}^{\mathcal{M},\lambda}$: 1) the Stochastic Hebb's Storage (SHS) which enables to realize an arbitrary optical operator, 2) the Stochastic Emergent Storage (SES) which instead aimed to the realization of optical memories.

### Stochastic Hebb's storage

First, we will employ this to realize an optical equivalent of the Hebbs rule: the *stochastic Hebbs storage* (SHS). Then we will show how the storage and recognition performance is greatly improved if SES is exploited.

With the SHS we want to generate a synaptic optical matrix $\mathbf{J}_\mathbf{T}^{\mathcal{M},\lambda}$ equivalent to an Hebb's matrix $\mathbf{T}$ with the aim to store the pattern $\boldsymbol{\phi}^*$. To do this, we rely on a linear combination of a set $\mathcal{M} = \{\mathbf{V}^1, \mathbf{V}^2 \dots \mathbf{V}^M\}$ of random optical synaptic matrices resulting from uncontrolled scattering:

$$\mathbf{J}_\mathbf{T}^{\mathcal{M},\lambda} = \sum_\nu^M \lambda^\nu \mathbf{V}^\nu \qquad (6)$$

Thus given Eq. (5), the *transformed intensity* $I_\mathbf{T}^{\mathcal{M},\lambda}(\boldsymbol{\phi})$ with the optical operator $\mathbf{J}_\mathbf{T}^{\mathcal{M},\lambda}$ emulates the Hamiltonian function associated to Hebb's synaptic matrix $\mathbf{T}$. Indeed the matrix $\mathbf{J}_\mathbf{T}^{\mathcal{M},\lambda}$ is connected to the intensities of the modes pertaining to the set $\mathcal{M}$ with the following equation:

$$\boldsymbol{\phi} \cdot \mathbf{J}_\mathbf{T}^{\mathcal{M},\lambda} \cdot \boldsymbol{\phi}^\dagger = \sum_\nu^M \lambda^\nu I^\nu(\boldsymbol{\phi}) = I_\mathbf{T}^{\mathcal{M},\lambda}(\boldsymbol{\phi}). \qquad (7)$$

The values for coefficients $\lambda^\nu$ are obtained by a Monte Carlo algorithm, (see methods) minimizing the difference between the target matrix and $\mathbf{J}_\mathbf{T}^{\mathcal{M},\lambda}$. Each coefficient may be then realized in hardware (mode-specific neutral density filters) or software fashion.

Employing SHS we can design any arbitrary optical operator if the two following ingredients are available: i) the access to the intensity $I^\nu(\boldsymbol{\phi})$ produced by a sufficiently large number of modes and ii) the correspondent optical synaptic matrix $\mathbf{V}^\nu$ for each mode. This is now possible with the Complete Couplings Mapping Method (CCMM, see methods), which enables the measurement of the intrinsic (no interference with a reference) $\mathbf{V}^\nu$ with a Digital Micromirror Device (DMD).

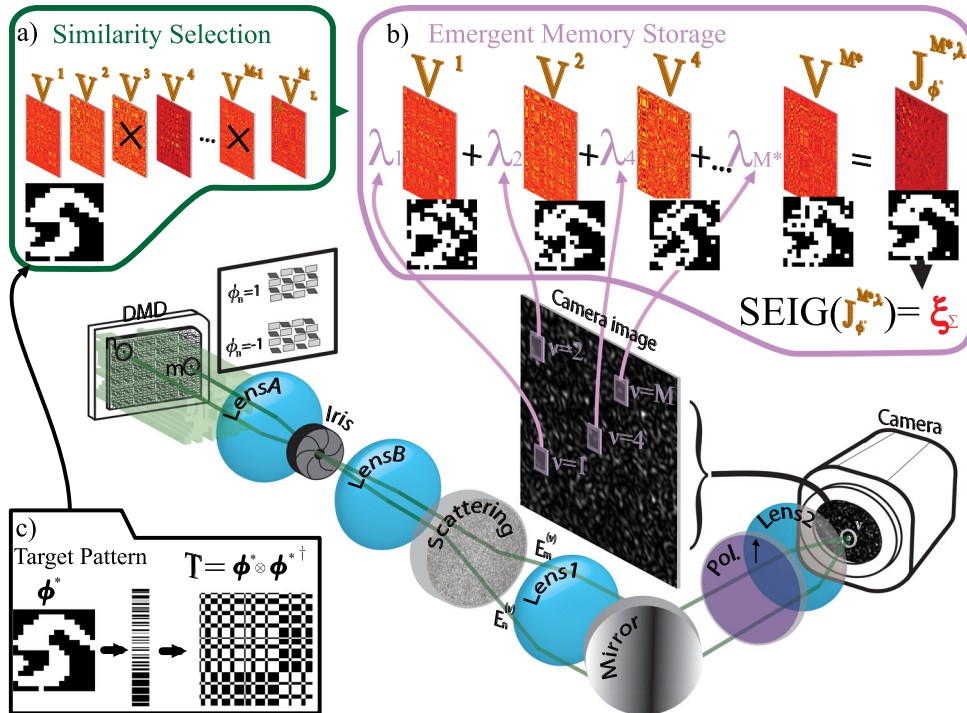

**Fig. 1 | Emergent memory storage scheme.** The sketch describes both the input query $\phi$ presentation and the measurement of the optical synaptic matrix $V^\nu$ (details in the Methods section). The first set of lenses (A-B) demagnifies (by a factor of 0.3) the DMD image, accommodating the scrambled input pattern of the scattering medium within Lens1's field of view. The second set of lenses (1-2) images the opaque medium's backplane onto the camera plane, with a magnification (factor of 11) that ensures the 1 speckle grain/mode per pixel imaging regime. **a**–**c** Illustrate the process of instantiating a memory in our architecture. **a** Represents the similarity selection stage, wherein optical synaptic matrix ($V^\nu$) are chosen based on their similarity with the target memory ($\phi^*$). **b** Illustrates the construction of an emergent memory through the summation of relative optical synaptic matrices ($\sum^M V^\nu = J_{\phi^*}^{\mathcal{M}^*,\lambda}$), resulting in the memory element $\xi_\Sigma$, obtained getting the largest eigenvector and computing the sign function (SEIG function). **c** Shows the pattern to be instantiated in memory $\phi^*$, its vectorization, and the corresponding coupling matrix constructed using Hebb's Rule (as employed in SHS).

With the CCMM, and the experimental apparatus shown in Fig. 1 see methods we are able to gather a repository $\mathcal{M}^L$ of tens of thousands ($M^t = 65536$) of optical synaptic matrices in minutes from which we sample a random subset $\mathcal{M}$ (with $M$ random samples) which we use as bases to construct our target artificial synaptic matrix.

The performance of this optical learning approach is shown in Fig. 2, in which we realized an Hebb's dyadic-like optical synaptic matrix (see insets of Fig. 2) from a ZnO scattering layer.

The memory pattern stored in our system is $\xi_\Sigma = \text{SEIG}(J_T^{\mathcal{M},\lambda})$, with SEIG(**H**) the operator that finds the eigenvector correspondent to the largest eigenvalue of **H** and then produces a binary vector with its elements' sign. Performance in storage and recognition for SHS are reported respectively in Fig. 2a, b (see methods). There we report the *Storage Error Probability* (the lower the better, indicates the average number of pixels differing between the stored and the target pattern, full definition in the, methods) and *Recognition Error* (the lower the better, the percentage of wrongly recognized memory elements out of a repository of 5000 presented patterns, full definition in the, methods).

SHS is basis hungry, requiring a large number of random optical synaptic matrices (which means modes/sensors/pixels) to successfully construct a memory element. his is connected to the fact that the target matrix T is constructed on $N \times N/2$ parameters (is symmetrical) acting as constraints, while we have $M$ free parameters to emulate it. A full emulation of $T$ is expected thus to be successful for $M > N \times N/2$ which is consistent with what we retrieve in Fig. 2 (Data for $M = 4096$ are out of scale as storage and recognition error is negligible).

**Stochastic emergent storage**

For the remainder of the paper, we will discuss how the performance drastically improves with SES. We recognize that each optical synaptic matrix contains the strongest of two memories $\xi^\nu = \text{SEIG}(V^\nu)$ then (instead of randomly extracting modes) we perform a similarity selection (see Fig. 1a and Supplementary Figs. 1 and 2 in the supplementary information file) in which we extract a set $M^*$ whose intrinsic memories are the closest possible to the target pattern $\phi^*$ (see insets of Fig. 1). The fact that in a mesoscopic laser scattering process, billions of independent modes can be produced and millions of them can be measured at once with modern cameras, is strategically employed in SES to boost the performance.

To perform the similarity selection with the optical modes the target pattern $\phi^*$ is compared with the eigenvectors of all the modes in the repository of characterized modes $\mathcal{M}^L$. The comparison is driven by the parameter $\mathcal{S}^\nu$

$$\mathcal{S}^\nu = \hat{\phi}^* \cdot \hat{\xi}^\nu \qquad (8)$$

that quantifies the degree of similarity between the first eigenvector of mode $\nu$, $\xi^\nu$, and $\phi^*$.

The modes $\nu$ providing the higher $\mathcal{S}^\nu$ are selected to feed a restricted repository of modes $\mathcal{M}^*$. The correspondent eigenvectors $\xi^\nu$ can be seen as prototypes of the target archetype, i.e. imperfect representations of the pattern to be stored (such as the one in Fig. 1c).

In SES, these prototypes interact constructively, generating a representation of the memory $\phi^*$ in an emergent fashion[14]. The interaction is obtained by the incoherent sum of the intensity of several pixels/modes with proper attenuation coefficients/weights $\lambda$.

The attenuation coefficients $\lambda$ are found by minimizing the distance between the archetype pattern to be stored $\phi^*$ and the matrix first eigenvector SEIG $(J_{\phi^*}^{\mathcal{M}^*,\lambda}) = \xi_\Sigma$ (see methods). Thus

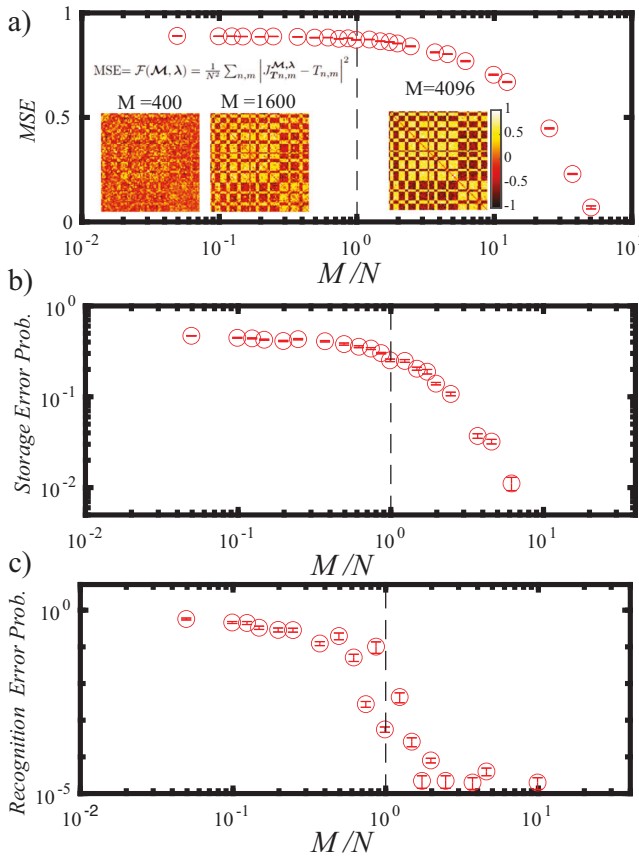

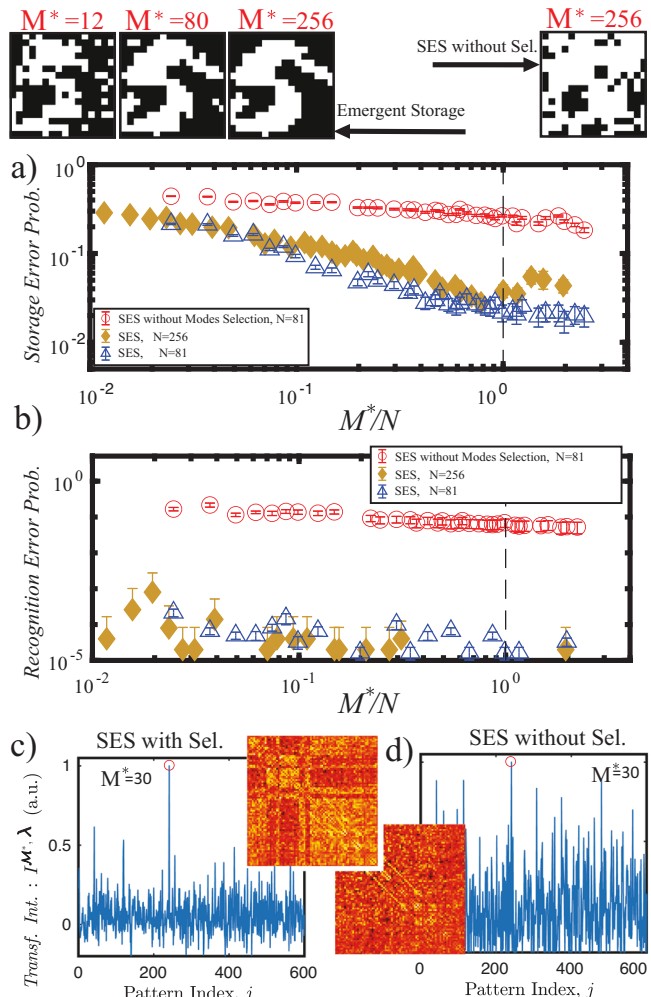

**Fig. 2 | Photonic Stochastic Hebb's Storage.** SHS realizes arbitrary optical operators emulating the target matrix **T** that stores the pattern $\boldsymbol{\phi}^{*}$ (as exemplified in Fig 1c). Our experimental setup utilizes $9 \times 9$ binary patterns ($N = 81$, $M^{*} = 65536$) and $M$ modes or camera pixels. **a** Presents the Mean Squared Difference (MSD) between the target and probe artificial optical synaptic matrices plotted against $M/N$. In **b**, we show the Storage Error Probability versus $M/N$. Finally, in **c**, we illustrate the Recognition Error Probability as a function of $M/N$. The insets within the figure display images of the reconstructed $\mathbf{J}_T^{\mathcal{M},\lambda}$ matrix for various values of $M$. Note that the point at $M/N \sim 50$ is out of scale because of a very low error rate. In all panels error bar represent standard error, obtained realizing 10 different target matrices **T** for each $M/N$ value, and measuring standard deviation $\sigma$ for each dataset and calculating standard error as $ERR = \sigma / \sqrt{(10 - 1)}$.

**Fig. 3 | Photonic Stocastic Emergent storage.** Top panels show the stored patterns obtained with SES for different values of $M^{*}$ with Similarity selection (three patterns on the left), and with random selection (pattern on the right). **a** Shows *Storage Error Probability* while **b** the *Recognition error probability*. Both with respect to $M^{*}/N$. **c** Transformed intensity for 600 patterns in the repository (pattern index $j$ on the ordinate axis) for the SES with the similarity selection (Sel.) stage and SES without the similarity selection. The mode $j = 241$, (with red circled intensity), correspond to the stored (recognized) pattern. **d** same as **c** but without similarity selection. The insets between **c** and **d** report the obtained $\mathbf{J}_{\boldsymbol{\phi}^{*}}^{M^{*},\lambda}$. Error bars in **a** and **b** are constructed as in Fig. 2.

substituting in Eq. (5) $\mathbf{J}_{\boldsymbol{\phi}^{*}}^{M^{*},\lambda}$ the transformed intensity in SES reads :

$$I_{\boldsymbol{\phi}^{*}}^{M^{*},\lambda}(\boldsymbol{\phi}) = \boldsymbol{\phi} \cdot \mathbf{J}_{\boldsymbol{\phi}^{*}}^{M^{*},\lambda} \cdot \boldsymbol{\phi}^{\dagger} = \sum_{\nu}^{M^{*}} \lambda^{\nu} I^{\nu}(\boldsymbol{\phi}). \tag{9}$$

The potential of SES is clarified in Fig. 3: the panels on the top left represent the stored pattern (target pattern is reported in Fig. 1c) for various sizes $M^{*}$ of the restricted repository. Note that SES greatly outperforms the random selection approach where emergent storage is absent (panel on the right).

Figure 3a shows the storage capability of the system. Blue triangles are relative to patterns with $N = 81$ elements, while for golden diamonds $N = 256$. The *Storage Error Probability* (Fig. 3c) improves more than an order of magnitude with respect to random selection (red circles). *Recognition Error Probability* (Fig. 3b) is three to four orders of magnitude better with respect to the randomly selected database. Note that the SES enormously outperforms SHS, indeed it is possible to perform recognition in the $M \ll N$ configuration, i.e. employing a number of camera pixels($M$) much smaller than the elements composing the pattern $N$.

Thus, in summary, SHS enables to create an optical operator of arbitrary nature, which can effectively execute diverse tasks. This versatility arises from its capability to construct an artificial optical synaptic matrix designed by the user, effectively emulating a matricial operator T. Conversely, SES focuses its functionality on generating an operator designed primarily for memory storage, excelling in this singular aspect. Consequently, it demands significantly less computational power and a smaller optical hardware setup (with a smaller $M^{*}$, see below), and enables lossy data compression (see supplementary information file and Supplementary Fig. 3).

This distinction influences the optimization procedure: SHS optimization relies on distances between matrices (measuring such distance computational cost scales as $N \times N$), while SES optimization is driven by distances between vectors (measuring such distance computational cost scales as $N$). Secondly, SES leverages preliminary similarity selection to identify the most relevant pixels/modes, a feature absent in SHS. As a result, the modes chosen for SES provide higher contrast in the classification task, especially in the $M < N$ regime.

In contrast, in the $M > N$ regime (more degrees of freedom than constraints), both approaches achieve essentially the same level of efficiency.

Figure 3c, d shows a recognition process example. The emergent learning process has been employed to store the pattern $\phi$ with index $j = 241$ from a repository of 5000 patterns. Figure 3c reports *transformed intensity* for the first 600 repository elements: a clear peak is distinguishable at $j = 241$ this implies that the pattern is recognized). The same graph is shown for the case of the random basis case (no similarity selection), in which recognition is more noisy.

**Photonic disordered classifier**

Our *disordered classifier* can work in parallel, simultaneously comparing an input with all memories stored, effectively working as a content addressable memory[4].

The experimentally retrieved *transformed intensity* for 4096 different memory elements $\phi^*$ is reported in Fig. 4a (organized in a camera-like $64 \times 64$ pixels diagram) for the proposed pattern $\phi$. Each value of $I_{\phi^*}^{M,\lambda}(\phi)$ represent the degree of similitude of $\phi$ to $\phi^*$. The patterns to the right side of the panel report the proposed pattern $\phi$ and the stored patterns relative to each arrow-indicated pixel. The pixel indicated with a red circle contains the pattern most similar to $\phi$ thus as expected produces the highest intensity. The system effectively works as a CAM in which an input query $\phi$ is tested in parallel against a list of stored patterns (the $\phi^*$) identifying the matching memory as the most intense *transformed intensity* pixel.

The interplay between Hopfield networks and Deep learning has been recently proposed and investigated[27,28]. In this framework here we demonstrate a new approach to perform higher rank categorical classification employing the cashed memories as features[29,30]: the deep-SES. We tested it on a 4500 randomly tilted digits images repository which is organized into 9 categories (digits from 1 to 9). We stored 3969 patterns/features in the disordered classifier ($m = 441$ per each digit), leaving 59 patterns per category for validation. In the camera-like diagrams (Fig. 4b, d) each category is found in the correspondent quadrant of the image. The two panels show the response of the disordered classifier to the inputs on the left for which the correspondent quadrants show a high number of intense pixels. Figure 4c shows integrated intensity after threshold. Figure 4e reports the confusion matrix for all labels, demonstrating categorical recognition efficiency above 90% which eventually may be enhanced employing error correction algorithms[31]. This result demonstrates the possibility to generate deeper optical machine learning achitectures and perform training by simply grouping memories. The potential of Deep-SES is further demonstrated by Fig. 4f, where we report a figure of merit comparing the efficiency of Deep-SES with Ridge Regression with Speckles (RRS)[21] (simulated). Note, while Deep-SES reaches an efficiency 90% for $M^* = 40$, the RRS suprasses this threshold for $M = 1600$. As $M^*$ represent the number of physical camera pixel employed in the classification, SES is capable of delivering a classifier with a much smaller hardware and computational complexity. The origin of this advantage, emerges form the fact that our memory writing process, selects pixels/modes which are the most correlated with the pattern to be recognized thus outperform with respect to randomly chosen ones. Moreover deep-SES enables thus to reorganize memories into new classes (reshuffling of classes) with almost no computational cost, a task which typically requires a new training in standard digital or optical architectures (see methods and supplementary information file, "Comparison with other platforms" section and Supplemenatary Table 1).

## Discussion

In summary, the Stochastic Emergent Storage (SES) paradigm enables classification with a significantly smaller number of sensors/pixels/modes compared to the elements composing the pattern. This opens up the possibility of fabricating complex pattern classifiers with

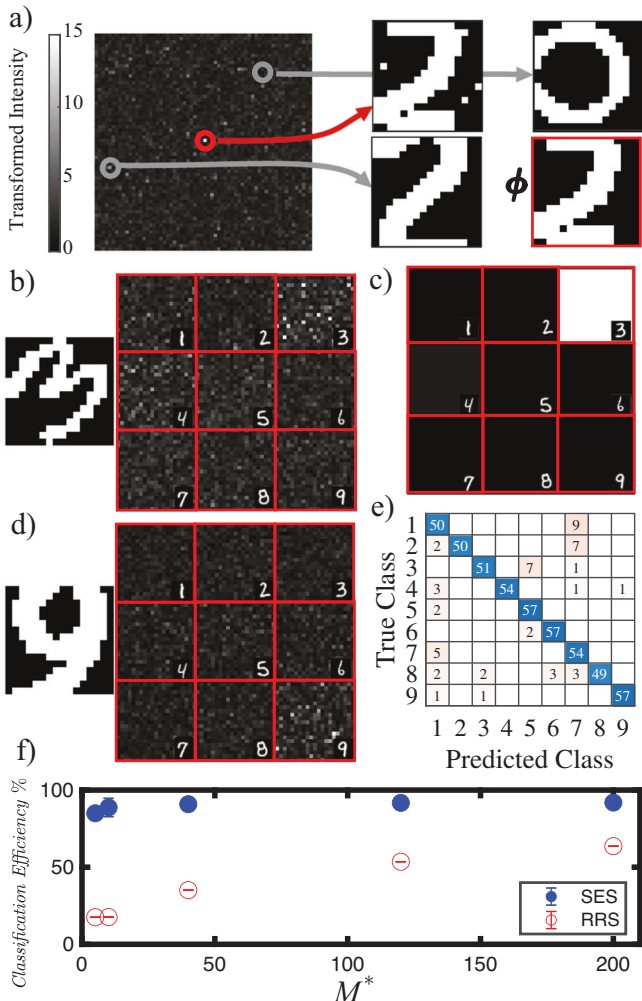

**Fig. 4 | Parallel/Categorical photonic classification. a** Shows the *transformed intensities* relative to 4096 stored memories ($N = 256$, $M = 120$) organized in $64 \times 64$ camera-like diagram. The *transformed intensities* are all generated as the pattern $\phi$ (shown on the right, highlighted by a red frame) is presented to the disordered classifier. For the circled pixels, the correspondent stored memory patterns are shown on the right (indicated by the arrow). The red circled pixel is associated to the memory which is the most similar to the pattern presented with the DMD. The diagram in **b** is similar to the previous but memories associated with 9 numerical categories, are organized in quadrants. The presented pattern $\phi$ is the "three" on the left. **c** Shows the thresholded and integrated intensity corresponding to the measure in **b**. **d** Is the same as **b** but with a "nine" pattern presented as input. **e** Reports the confusion matrix for the categorical (number class) recognition. **f** Reports the classification efficiency on the same digit database for Deep-SES and the Ridge Regression with Speckles (RRS) (for $M = 120$ recognition efficiency = $91.71\% \pm 0.8$ (95% confidence interval, size of the statistics = 59))[21], versus $M$ (see Methods). Error bars in **f** are constructed as in Fig. 2.

only a few detecting elements, eliminating the fabrication processes. Deep-SES offers a new paradigm for network training, enabling to generate classes just by grouping memories, and it opens the way to a computation-free rearrangement of classes.

The paradigms presented here can be potentially exported to other disordered systems, such as biological neural networks or neuromorphic computer architectures while exploring the emergent learning process in these systems can also provide valuable insights into the memory formation process in the brain.

The results presented in this study contributes to the ongoing challenge of understanding the biological memory formation process. There are currently two major hypotheses that are the subject of

debate, the connectionist hypothesis[5], which suggests that neural networks form new links or adjust existing ones when storing new patterns, and the innate hypothesis[32], which posits that patterns are stored using pre-existing neural assemblies with fixed connectivity. One central aspect in this ongoing debate pertains to the 'efficiency' of the network, a facet that, in both artificial and natural networks, immediately invokes considerations related to energy consumption. On one side, it has long been established that in Hebbian networks, the number of memories (W) scales linearly with the number of nodes (N), expressed as $W = \alpha N$. For this reason, many research efforts are dedicated to optimizing the proportionality constant $\alpha$, which however appears to be upper bounded to two. On the other side, it has been recently demonstrated, both numerically[7,9] and theoretically[33,34], that in the stochastic innate approach, the number of memory increases exponentially with the number of nodes: $W \propto e^{aN}$. In other words, for larger system sizes, the innate approach predicts a significantly greater number of memories compared to the connectionist perspective. The "complexity" of the system (artificial neural network or brain), denoted as $S = \lim_{N\to\infty} \log(W)/N$, tends to zero for the connectionists, whereas it remains non-zero for the innatists.

SES introduces a fresh perspective to the problem by leveraging the Hebbian structure of the synaptic matrix, with a foundation of the connectionist hypothesis. However, SES goes beyond by exploring the potential of a stochastic innate network in which, pre-existent random synaptic structures are combined to generate memory elements in an emergent manner. Whether the SES could bring a new point of view, lumping together the innatism and connectivism, is a fascinating hypothesis, that must be explored in the future.

## Methods
### Background
In our experiment (similarly to a typical wavefront shaping experiment), light from a coherent source is controlled by a spatial light modulator and transmitted after propagation through a disordered medium into the mode $v$. The field transmitted at $v$ is described as

$$E^\nu = \sum_{n=1}^{N} E_n^\nu \phi_n \tag{10}$$

where the index $n$ runs on the controlled segments at the input of the disordered medium, $E_n^\nu$ is the field resulting from laser field from the $n$th segment transformed by the transmission matrix element on the sensor $v$ and $\phi_n^\nu$ is the phase value from the wavefront modulator. In our experiment, we consider the simplified configuration in which $\phi_n^\nu \in \{-1, +1\}$.

The field at $v$ can be separated in its two components: the field-at-the-segment $A_n$ and transmission matrix element $t_n^\nu$

$$E_n^\nu = A_n t_n^\nu \tag{11}$$

Indeed the $E_n^\nu$ are Gaussianly distributed complex numbers:

$$E_n^\nu = \xi_n^\nu + i\eta_n^\nu. \tag{12}$$

In the case in which just two segments $n$ and $m$ are active and in the $+1$ configuration, we can ignore the $\phi_n$:

$$E^\nu = E_n^\nu + E_m^\nu = \xi_n^\nu + i\eta_n^\nu + \xi_m^\nu + i\eta_m^\nu. \tag{13}$$

In absence of modulation, intensity is written as the modulus square of $E^\nu$

$$\begin{aligned} I^\nu &= |E_n^\nu + E_m^\nu||E_n^{\nu\dagger} + E_m^{\nu\dagger}| \\ &= |E_n^\nu|^2 + |E_m^\nu|^2 + |E_n^\nu||E_m^{\nu\dagger}| + |E_n^{\nu\dagger}||E_m^\nu|. \end{aligned} \tag{14}$$

we recognize that

$$|E_n^\nu||E_m^{\nu\dagger}| = \xi_n^\nu\xi_m^\nu - i\xi_n^\nu\eta_m^\nu + i\eta_n^\nu\xi_m^\nu + \eta_n^\nu\eta_m^\nu \tag{15}$$

$$|E_n^{\nu\dagger}||E_m^\nu| = \xi_n^\nu\xi_m^\nu + i\xi_n^\nu\eta_m^\nu - i\eta_n^\nu\xi_m^\nu + \eta_n^\nu\eta_m^\nu \tag{16}$$

$$|E_n^\nu||E_m^{\nu\dagger}| + |E_n^{\nu\dagger}||E_m^\nu| = 2\xi_n^\nu\xi_m^\nu + 2\eta_n^\nu\eta_m^\nu. \tag{17}$$

thus

$$I^\nu = \xi_n^{\nu 2} + \eta_n^{\nu 2} + \xi_m^{\nu 2} + \eta_m^{\nu 2} + 2\xi_n^\nu\xi_m^\nu + 2\eta_n^\nu\eta_m^\nu. \tag{18}$$

or

$$I^\nu = E_n^{\nu 2} + E_m^{\nu 2} + 2\xi_n^\nu\xi_m^\nu + 2\eta_n^\nu\eta_m^\nu. \tag{19}$$

In general for $N$ segments in an arbitrary configuration of the modulator

$$I^\nu = \sum_{n,m}^{N} E_n^{\nu 2} + E_m^{\nu 2} + 2(\xi_n^\nu\xi_m^\nu + \eta_n^\nu\eta_m^\nu)\phi_n\phi_m. \tag{20}$$

the argument of the sum can be written in matrix form defining the matrix $\mathbf{V}^\nu$ also named optical coupling matrix:

$$V_{nn}^\nu = E_n^{\nu 2} = \xi_n^{\nu 2} + \eta_n^{\nu 2} \tag{21}$$

$$V_{nm}^\nu = \xi_n^\nu\xi_m^\nu + \eta_n^\nu\eta_m^\nu \tag{22}$$

Matrix $\mathbf{V}^\nu$ is a bi-dyadic matrix and it can be rewritten in matricial notation:

$$\mathbf{V}^\nu = \boldsymbol{\xi}^\nu \otimes \boldsymbol{\xi}^{\nu\dagger} + \boldsymbol{\eta}^\nu \otimes \boldsymbol{\eta}^{\nu\dagger} \tag{23}$$

where the **notation** indicates a vector on lowercase Greek letters and a matrix on uppercase, while $\dagger$ is the conjugate transpose operator. Being bi-dyadic the matrix possesses the eigenvectors $\boldsymbol{\xi}^\nu$ and $\boldsymbol{\eta}^\nu$ by construction. Note that $\boldsymbol{\xi}^\nu, \boldsymbol{\eta}^\nu \in \mathbb{C}^N$, $\mathbf{V}^\nu \in \mathbb{C}^{N\times N}$ and is Hermitian.

When modulation is present with an input modulation pattern $\boldsymbol{\phi}$

$$I^\nu(\boldsymbol{\phi}) = \sum_{n,m}^{N} V_{nm}^\nu \phi_n \phi_m = \boldsymbol{\phi} \cdot \mathbf{V}^\nu \cdot \boldsymbol{\phi}^\dagger \tag{24}$$

Note that even if $\mathbf{V}^\nu$ is a complex matrix, being Hermitian, the double scalar product produces a real scalar because inverted sign imaginary contributions from above and below the diagonal result eliminated reciprocally thus producing a positive real intensity. The optical operator $\mathbf{V}^\nu$, associates thus the pattern/array $\boldsymbol{\phi}$ to the scalar $I^\nu$ which is a measure of the degree of similitude of $\boldsymbol{\phi}$ to the first eigenvector of $\mathbf{V}^\nu$, $\mathrm{EIG}(\mathbf{V}^\nu) = \boldsymbol{\xi}^\nu$.

Note that to simplify the realization of the experiment, we operate in the configuration in which each mode $v$ corresponds to a single sensor. As we employ a camera to measure $I^\nu$, e the one-mode-per-pixel configuration is obtained by properly tuning the optical magnification.

### Stochastic Hebb's storage protocols details
By summing intensity measured at two modes $v_1$ and $v_2$, and considering linearity of the process:

$$\begin{aligned} I^{\nu_1} + I^{\nu_2} &= \boldsymbol{\phi} \cdot \mathbf{V}^{\nu_1} \cdot \boldsymbol{\phi}^\dagger + \boldsymbol{\phi} \cdot \mathbf{V}^{\nu_2} \cdot \boldsymbol{\phi}^\dagger = \\ &= \boldsymbol{\phi} \cdot \mathbf{J}^{\nu_1, \nu_2} \cdot \boldsymbol{\phi}^\dagger. \end{aligned} \tag{25}$$

Generalizing, i.e. summing intensity at an arbitrary number $M$ of modes pertaining to the set $\mathcal{M} = \{\nu_1, \nu_2, \ldots \nu_M\}$, we retrieve

$$I^{\mathcal{M}} = \sum_{\nu}^{M} \boldsymbol{\phi} \cdot \mathbf{V}^{\nu} \cdot \boldsymbol{\phi}^{\dagger} = \boldsymbol{\phi} \cdot \mathbf{J}^{\mathcal{M}} \cdot \boldsymbol{\phi}^{\dagger} \qquad (26)$$

with

$$J_{nm}^{\mathcal{M}} = \sum_{\nu}^{M} V_{nm}^{\nu} \qquad (27)$$

Thus, the optical operator textbf$\mathbf{J}^{\mathcal{M}}$ associates a pattern/array $\boldsymbol{\phi}$ to the scalar $I^{\mathcal{M}}$, the *transformed intensity*, which is a proxy of the degree of similitude of $\boldsymbol{\phi}$ to the first eigenvector of $\mathbf{J}^{\mathcal{M}}$: $\mathrm{EIG}(\mathbf{J}^{\mathcal{M}}) = \boldsymbol{\xi}_{\mathcal{M}}$. To deliver an user-designed arbitrary optical operator, we introduce the tailored attenuation coefficients $\lambda^{\nu} \in [0, 1]$. These can be both obtained in "software version" (multiplying each $I^{\nu}$ by an attenuation coefficient $\lambda^{\nu}$) or by realizing a mode-specific hardware optical attenuator (such as proposed in the sketch in Fig. 1, fuchsia windows, see below).

*Transformed intensity* with the addition of the attenuation coefficients reads as:

$$I^{\mathcal{M},\boldsymbol{\lambda}} = \sum_{\nu}^{M} \boldsymbol{\phi} \cdot \lambda^{\nu} \mathbf{V}^{\nu} \cdot \boldsymbol{\phi}^{\dagger} = \boldsymbol{\phi} \cdot \mathbf{J}^{\mathcal{M},\boldsymbol{\lambda}} \cdot \boldsymbol{\phi}^{\dagger} \qquad (28)$$

In SHS, the absorption coefficients $\boldsymbol{\lambda}$ are the free parameters which enable to design the arbitrary optical operator $\mathbf{J}^{\mathcal{M},\boldsymbol{\lambda}}$. For example, to replicate the dyadic matrix constructed with he Hebb's rule $\mathbf{T}$ and capable to store the pattern $\boldsymbol{\phi}$ (see Fig 1c of the main paper) one has to select $\boldsymbol{\lambda}$ so that the function

$$\mathcal{F}(\mathcal{M}, \boldsymbol{\lambda}) = \sum_{n,m}^{N} \left| \sum_{\nu}^{M} \lambda^{\nu} V_{nm}^{\nu} - T_{nm} \right|^2 = \\ = \mathrm{DIST}\left(\mathbf{J}^{M,\lambda}, \mathbf{T}\right) \qquad (29)$$

is minimized. We name $\mathbf{J}_{\mathbf{T}}^{\boldsymbol{M},\boldsymbol{\lambda}}$ the artificial optical synaptic matrix in which $\boldsymbol{\lambda}$ have been optimized to deliver the optical operator $\mathbf{T}$, and

$$I_{\mathbf{T}}^{\boldsymbol{M},\lambda} = \sum_{\nu}^{M} \lambda^{\nu} I^{\nu} \qquad (30)$$

the relative *transformed intensity*.

This approach employs the random, naturally-occurring optical synaptic matrices from the set $\mathcal{M}$ as a random basis on which to build the target optical operator. Its effectiveness is thus dependent on the number of free parameters with respect to the constraints. The constraints are the number of independent elements that have to be tailored on $\mathbf{T}$. These are $\Pi = (N(N-1)/2$ as $\mathbf{T}$ is symmetric. Indeed as shown in Fig. 2 of the main paper (inset of panel b) for the $N = 81$ case, it is possible to replicate almost identically $\mathbf{T}$ when $M > \Pi$, that is when the number of free parameters (the $\boldsymbol{\lambda}$) is comparable with the constraints.

## Storage error probability

In our storage paradigm, the stored pattern corresponds to the eigenvector of the $\mathbf{T}$. As we are employing binary patterns, the sign operation is needed. The stored pattern is thus $\mathrm{SEIG}(\mathbf{J}_{\boldsymbol{\phi}^*}^{\mathcal{M},\boldsymbol{\lambda}}) = \boldsymbol{\xi}_{\Sigma}$, where the $SEIG()$ operator retrieves the first eigenvalue of a matrix and applies the sign operation to it. The *Storage Error Probability* reported in Figs. 3 and 2 the storage process effectiveness. First, we calculate the number of elements of $\boldsymbol{\xi}_{\Sigma}$ which differ from the target memory $\boldsymbol{\phi}^*$, $S\_ERR$. The value of $S\_ERR$ can be seen as the number of error pixels in the stored pattern image.

Then we compute

$$Storage\,Error\,Probability = S\_ERR/N. \qquad (31)$$

For storage purposes, obviously the lower, the *Storage Error Probability* the better.

## Recognition error probability

The optical operator $\mathbf{J}_{\boldsymbol{T}}^{\boldsymbol{M},\lambda}$ associates the *transformed intensity* scalar to each input pattern $\boldsymbol{\phi}$:

$$I_{\mathbf{T}}^{\mathcal{M},\boldsymbol{\lambda}} = \boldsymbol{\phi} \cdot \mathbf{J}_{\mathbf{T}}^{\mathcal{M},\boldsymbol{\lambda}} \cdot \boldsymbol{\phi}^{\dagger}. \qquad (32)$$

we can thus employ the experimentally measured *transformed intensity* to recognize patterns. We employed a repository of $P = 5000$ patterns containing digits with random orientation (https://it. mathworks.com/help/deeplearning/ug/data-sets-for-deep-learning. html), labeling as recognized patterns, the ones producing a transformed intensity above 10 standard deviations from the values obtained probing randomly generated binary patterns. The value $R\_ERR$ is the number of wrongly identified patterns experimentally.

Indeed, the *transformed intensity* is obtained experimentally optically presenting the pattern to our disordered classifier. The step-by-step presentation procedure is the following: *i)* the probe pattern $\phi$ is printed onto a propagating laser beam employing a DMD in binary phase modulation mode (see experimental section), *ii)* light scattered by the disordered medium is retrieved for the relevant mode/pixel set $\mathcal{M}$, *iii)* the transformed intensity measured by the selected sensors/camera pixels is obtained with Eq. (30), *iv)* a pattern is defined as recognized if the *trained transformed intensity* results higher than the threshold. The *Recognition Error Probability* is then obtained as

$$Recognition\,Error\,Probability = R\_ERR/P. \qquad (33)$$

The Supplementary Fig. 4 visualizes for the classification/recognition process.

Note that Storage Error Probability ($S\_ERR$) and Recognition error probability ($R\_ERR$) provide insights on two very different aspects of our storage platform performance. $S\_ERR$ is essentially a storage fidelity observable, counting the ratio of wrong/correct pixels in the pattern to be stored which differ from the target memory to be stored $\boldsymbol{\phi}^*$, and accounts for the efficiency of our approach (the emergent storage) to instantiate a target memory in a memory repository. $R\_ERR$ retrieves recognition efficiency, thus reports on the ratio of memory retrieval tests providing wrong memory addresses, when different input patterns from a repository are proposed as stimuli. The $S\_ERR$ influences $R\_ERR$: i.e. if many error are present in the pattern injected in a repository the recognition fails. However $R\_ERR$ is also affected by other features such as for example the order of nolinearity (we use intensity do appreciate differences in the field thus we employ a second order nonlinearity) which influences the capability to differentiate similar patterns and also the structure of the repository (if the repository contains very similar patterns then the recognition task is more difficult). Thus the relation is not a simple proportionality, while the two observable look at two very different aspects of the memory process i.e. storage fidelity and recognition efficiency.

In Deep-SES instead, a single probe pattern is compared with many memories. We performed this task with digital data analysis but all the processes can be realized analogically, by performing pixel selection and weighting with DMDs. In such a case the probe pattern is directly tested against many memories: all the ones composing the training set. For the 9 class digit classification reported in the Fig, 3969 individual memories (441 per class) have been used. Employing a DMD with 33 kHz frame rate would mean essentially performing optical classification in 0.1 seconds.

## Stochastic emergent storage protocol details

In SES (see code and data at[35]) we exploit the fact that any optical coupling matrix $\mathbf{V}^\nu$ is a bi-dyadic thus hosting two intrinsic but random patterns:

$$\mathbf{V}^\nu = \boldsymbol{\xi}^\nu \otimes \boldsymbol{\xi}^{\nu\dagger} + \boldsymbol{\eta}^\nu \otimes \boldsymbol{\eta}^{\nu\dagger} \tag{34}$$

thus the optical coupling matrix at location $\nu$, $\mathbf{V}^\nu$, hosts the two random memory vectors $\boldsymbol{\xi}^\nu$ and $\boldsymbol{\eta}^\nu$.

To employ these disorder-embedded structures as memories we resorted to the following multi step strategy.

i. We start measuring the transmission matrices from a large set of modes employing the Complete Couplings Mapping Method (CCMM, see below). We monitor $M^L = 65536$ modes employing a region of interest for the camera of $256 \times 256$ pixels in the one-mode-per-pixel configuration. The retrieved transmission matrices are saved into a computer memory and compose our starting random structures repository $\mathcal{M}^L$.

ii. We computationally find the first eigenvector $\boldsymbol{\xi}^\nu$ for each measured matrix $\mathbf{V}^\nu$

iii. The user, designs a target memory pattern to be stored $\boldsymbol{\phi}^*$ and a number $M^*$ of modes to be employed.

iv. The target pattern $\boldsymbol{\phi}^*$ is compared with all the eigenvectors in $\mathcal{M}^L$ by computing the similitude degree $\mathcal{S}$:

$$\mathcal{S}^\nu = \hat{\boldsymbol{\phi}}^* \cdot \hat{\boldsymbol{\xi}}^\nu \tag{35}$$

with the symbol $\hat{i}$ indicating vector normalization: $\hat{i} \cdot \hat{i} = 1$.

v. The set of modes $\mathcal{M}^L$ is similarity-decimated to the set $\mathcal{M}^*$, i.e. we select the $M^*$ modes with the higher $\mathcal{S}^\nu$ values to be part of the new, reduced repository $\mathcal{M}^*$.

Once $\mathcal{M}^*$ is realized, we need to "train" the attenuation coefficients $\boldsymbol{\lambda}$. The attenuation values are selected between 16 values degrees of absorption in the $\in [0, 1]$ range, so that they are identified with a 4 bits number.

After initializing the lambda and computing the initial configuration optical operator

$$\mathbf{J}^{\mathcal{M}^*, \boldsymbol{\lambda}} = \sum_{\nu}^{M^*} \lambda^\nu \mathbf{J}^\nu \tag{36}$$

the $\boldsymbol{\lambda}$ are optimized with a Monte Carlo algorithm. At each optimization step a single $\lambda^\nu$ is modified and the change is accepted if the eigenvector similarity function

$$\mathcal{F}^*(\boldsymbol{\lambda}, \boldsymbol{\phi}^*, \mathcal{M}^*) = \hat{\boldsymbol{\xi}}_\Sigma \cdot \hat{\boldsymbol{\phi}}^* \tag{37}$$

decreases. Note that in Eq. (37), $\boldsymbol{\xi}_\Sigma$ is the first eigenvector of $\mathbf{J}^{\mathcal{M}^*, \boldsymbol{\lambda}}$.

After a sufficiently large number of steps $\mathcal{F}^*(\boldsymbol{\lambda}, \boldsymbol{\phi}^*, \mathcal{M}^*)$ is minimized and form the final configuration of $\boldsymbol{\lambda}$ we obtain the final version of the optical operator: $\mathbf{J}_{\boldsymbol{\phi}^*}^{\mathcal{M}^*, \boldsymbol{\lambda}}$.

Note that the previous procedure can be cast in a computationally lighter version replacing some digital operations with optical measurements. The similarity selection can be substituted with intensity measurement. Indeed intensity itself is a direct measure (see Eq. (24)) of the degree of similarity of the probe pattern with the correspondent $\boldsymbol{i}^\nu$ vector, thus similarity selection can be substituted by an optical operation with cost $M^L$.

## Experimental setup and CCMM

The same experimental setup is employed for two tasks. The first is the measurement of the optical synaptic matrices $\mathbf{V}^\nu$, the second is to perform classification, presenting to the disordered classifier a

test pattern $\boldsymbol{\phi}$ and retrieving the *transformed intensities* for each trained memory. A sketch of the experimental setup is provided in Supplementary Fig. 1 in supplementary information file.

We employ a single mode laser (AzurLight 532, 0,5W) with beam to about 1 cm. Then it is fragmented into $N$ individually modulated light rays controlled by a Digital Micromirror Device (DMD)[36] composed by $1024 \times 768$ (Vialux, V-7000, pixel Pitch 13.68 μm, 22 kHz max frame rate) flipping mirrors which can be tuned into two configurations (on or off). Phase modulation is obtained employing the super-pixel method (see refs. 23,37) which require a spatial filtering to isolate the selected diffraction orders. DMD pixels are organized into $N$ 4-elements super-pixels (segments) capable to produce a 0 or $\pi$ phase pre-factors equivalent to field multiplication by $\phi_n = \in \{-1, 1\}$. The bundle of light rays is then scrambled by a diffusive, multiple scattering medium (60 μ layer of ZnO obtained from ZnO powder from Sigma Aldrich item 544906-50g, transport mean free path 8 μm[38]). The $N$ super-pixels are organized on the DMD in a square array, which is illuminated by an expanded laser Gaussian beam (diameter of about 1 cm). Indeed, the DMD surface is imaged onto the Diffusive medium (0.3 × de-magnification). This de-magnification is required to ensure the diffused image to fit into the selected detection camera ROI. Then, the back layer of the disordered structure is imaged on the detection camera (11 × magnification). This magnification has been chosen to minimize the speckle grain size in order to work in the one-mode-per-pixel configuration (one-pixel-per-speckle-grain). The optical collection apparatus, does not require a particular performance, indeed we employed a commercial, low-cost 25.45 mm focal bi-convex lens for the light collection from the far side of the sample. Several constraints have to be considered in the experimental design. When light from a DMD super-pixel emerges from the disordered medium, it is diffused into a larger disk-shaped area. For this reason, we have to ensure that each these light disks are interfering with all the disks generated by other super-ixels in the detection camera ROI, and this introduces a constraint on the maximum ROI size ($M^L$). The size of these diffusion disks is regulated by the thickness of the disordered scattering medium. Nevertheless, note that increasing the scattered thickness also decreases the light intensity on the camera and the stability of the speckle pattern thus a trade-off between thickness and signal-stability should be found at the experimental design step.

Superpixel method is obtained thanks to 2.66 mm aperture iris in front of the DMD. As shown in Eq. (19) when two DMD mirrors are activated:

$$I_{n,m}^\nu = E_n^{\nu 2} + E_m^{\nu 2} + 2\xi_n^\nu \xi_m^\nu + 2\eta_n^\nu \eta_m^\nu. \tag{38}$$

while if a single segment is activated

$$I_n^\nu = E_n^{\nu 2}. \tag{39}$$

Thus putting together Eq. (38) and Eq. (39) one obtains

$$V_{nm}^\nu = \xi_n^\nu \xi_m^\nu + \eta_n^\nu \eta_m^\nu = \frac{I_{nm}^\nu - I_n^\nu - I_m^\nu}{2}. \tag{40}$$

Thus to determine one single element of the optical synaptic matrix, one has to perform three intensity measurements. The total number of measurement to reconstruct the full synaptic matrix is $\Pi = N(N-1)/2$ (as $V_{nm}^\nu = V_{mn}^\nu$ i.e. the optical synaptic matrix is symmetric then just the above-the-diagonal elements need to be measured). For $N = 256$ this means that 32896 measurements are required, which can be obtained in maximum 5 minutes employing our DMD-Camera experimental setup (speed bottleneck from the camera sensor which works at ~150 frames per second). At each measurement we take a image from a Region Of Interest (ROI) of $256 \times 256$ pixels, thus collecting info for $M^L = 65536$ modes. This measurement realizes modes,

random memories and optical synaptic matrix for the database $\mathcal{M}^L$. Experimental data are organized into a $32896 \times 256 \times 256$ matrix. In our case increasing the size of $N$ or $M^t$ is limited by the size of the Random Access Memory size of the computing workstation.

### Satistics and reproducibility
In error bars in Figs. 2–4 represent standard error, obtained realizing 10 different target matrices **T** for each $M/N$ value, and measuring standard deviation $\sigma$ for each dataset and calculating standard error as $\text{ERR} = \sigma/\sqrt{(10-1)}$. **T** matrices where blindly and randomly extracted at each measure from a 5000 elements pattern repository. No statistical method was used to predetermine sample size. No data were excluded from the analyses.

## Data availability
Experimental and generated data related to the generated in this study are deposited in the GitHub repository at the address https://doi.org/10.5281/zenodo.10222344[35].

## Code availability
Code realized in this study are deposited in the GitHub repository at the address https://doi.org/10.5281/zenodo.10222344[35].

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

## Acknowledgements
This research was funded by: Regione Lazio, Project LOCALSCENT, Grant PROT. A0375-2020-36549, Call POR-FESR "Gruppi di Ricerca 2020" (to M.L.); ERC-2019-Synergy Grant (ASTRA, n. 855923; to GR); EIC-2022-PathfinderOpen (ivBM-4PAP, n. 101098989; to G.R.); Project "National Center for Gene Therapy and Drugs based on RNA Technology" (CN00000041) financed by NextGeneration EU PNRR MUR - M4C2 - Action 1.4 - Call "Potenziamento strutture di ricerca e creazione di

campioni nazionali di R&S" (CUP J33C22001130001) (to G.R.). The authors Acknowledge Enrico Ventura, and Luigi Loreti for fruitful discussions.

## Author contributions

M.L. Conceived the Idea, Designed and Realized the experiments, Analyzed the data. G.G. analyzed the data, developed the geometrical interpretation and performed numerical simulations of RRS. G.R. conceived the mapping with the innate and connectionist conjectures. All authors contributed to data interpretation and writing the manuscript.

## Competing interests

The authors declare no competing interests.
