## [Peer Review File · Nature Communications]

Photonic Stochastic Emergent Storage for deep classification by scattering-intrinsic patternsReviewer #1 (Remarks to the Author):

The author has utilized the naturally occurring random structure to store arbitrary patterns in this paper. In particular, in this work, the authors introduce the idea of stochastic emergent storage to overcome one of the challenges of the Hopfield network, namely, the capacity limit in the densely populated network which rises from the emergence of uncontrolled memory states.

The SES utilizes database for random patterns generated by a disorder and self-assembled structure such that a set of prototypes resembles the target memory via similarity criterion.

Next, the author also leverages unique properties of photonic computation such as wavefront transformation to implement the SES protocol experimentally without any additional fabrication.

More specifically, in this SES optical-based hardware, the optical transmission matrices allow one to perform both pattern storage and pattern retrieval in a way that an input can be compared to thousands of stored elements via a two-layer architecture leading to deep classification.

To provide detail on the implementation, the author provides basic knowledge of wavefront scattering and discusses how one can reconstruct the optical synaptic matrix. Below I list a few questions.

The main figures in the paper should be more comprehensive and explained in an easy way to convey the message directly without too much reference to the appendix or main text. In some sense, the figure should have enough captions that can be considered as complementary to the main text. I believe the author can improve the flow of figure 1 and its caption to make it more accessible. For example, it is not clear what is the scattering medium and what is the role of all different lenses.

Also, the similarity selection is an important part of the work and I guess, needs to be clarified in the main text.

The main quantities on the figure are required to be defined in the main text and the figure also requires more detailed discussion. More specifically, why the performance has different behavior in decreasing of storage vs classification. Naively, I would have thought the classification error for the same ratio of M/N should stay larger.

Maybe I am missing the point here, but I would appreciate if the author can elaborate on how in SES, one can impose the constructive interaction of the prototype.

In figure 3, again, would the author explain why the SES enhances the classification error by a few orders of magnitude better? Part c and d, require the information in the caption.

The author mentions that exploring the emergent learning process can provide insight into memory formation. This is indeed a very interesting point, maybe it could have been a bit elaborated. In particular, the connection of SES with connectionist hypothesis.

While I find the main results and effort of the authors very interesting and great potential publishing, however, I feel there is a need to improve the presentation and the flow of the work to make it accessible to readers, given the interdisciplinary nature of the work.

Reviewer #2 (Remarks to the Author):

The authors propose and demonstrate a new concept for optical deep classification exploiting the intrinsic disorder arising from a random diffuser scattering. They also pioneered a similarity-based algorithm to reduce the number of basis elements and, thus, reduce the computation. This is an interesting approach that gets rid of the complexity imposed on the diffractive elements in similar free-space optical approaches, and instead moves it to conventional computer-based post-processing tasks. This could translate into a simplification of the experimental setup to perform this type of tasks.

I am in principle supportive of this manuscript to be accepted for publication in Nature Communications. However, I have many questions for which I couldn't find answers, several details could use more in detail discussion (e.g., the limitations of the approach, the difference between SHS and SES, etc.), and there is no benchmarking with respect to similar approaches so a direct comparison renders difficult. I suggest a revision of the structure and discussion of the manuscript to address the following points. In particular, I suggest the authors to use subtitles to clearly separate the different sections and have clearer transitions, as found below, I'm particularly confused by Fig. 2 and Fig. 3 and the actual differences between SHS and SES.

- The number of measurements and computations needed in this approach come with a penalty in processing time, although this, which is basically "training," is time consuming regardless of the platform. How does this approach compare to others? Suggesting a figure of merit and comparing to other approaches can be useful to understand the real advantage of the method.
- What is the purpose of the 0.3 x de-magnifications? The image of each DMD pixel after demagnification is about half the mean free-path of the diffuser. Is there an intrinsic limitation for the method in terms of how these two parameters compare? That clearly determines the scattering, but how does that affect the technique and its precision?
- Continuing with the magnification, the second magnifying system is design to achieve 11x. The authors mention that this is to "minimize the speckle size". Wouldn't it maximize it if we think in terms of size (in a way that it fits entirely within the detector size) or what do the authors mean with minimize?
- I am confused about the selection of the random subset of (cursive) M out of the full ML in Fig. 2 for SHS. Why was it introduced in first place? Is the random subset vs. similarity selection the only difference between SHS and SES? I might have missed a big point here.
- Fig. 2: we know $N=81$, but what is N? N is somehow defined in the methods, but at this point in the main text the reader doesn't know what N is. Moreover, in the methods, the authors suggest $T=N \times N$ but later on, $N \times N$ is also the number of superpixels in the DMD, is this correct?.
- Fig. 2: M is the number of random samples, isn't it?. The meaning of M and N should be added to the caption.
- Fig. 2: there is not discussion of the meaning of these results in the main text. The authors simply show it but do not discuss what it means, for example, to achieve low error for $M > 10N$ or why $M=N$ (which is a significantly small number for M) yields lower classification error, like they did in Fig. 3. They also mention that SES is basis hungry, but do not elaborate on this statement. If $M=10*N$ with $N=81$ is enough to have low errors, then the basis is not that big after all if we consider that M is initially over 65k matrices.
- Fig. 2: the inset in Fig. 2b shows $M=4096$, however the data point $M/N \sim 50$ is not plotted. The max is 10. Why?
- If I am understanding correctly from Eq. (5) and Eq. (6), the results in Fig. 2 for random subsets of M with $N=81$ should be equivalent to SES for $M*$ random subsets. However, Fig. 3 shows, for the same N and a random basis, much worse results than in Fig. 2. What is different and why?
- What are the limitations of the approach in terms of spatial frequencies?
- The process to perform the classification is not fully clear to me. How fast is the classification? What are the experimental and post processing components? I would appreciate a similar figure like Fig. 1 for the storage scheme, but with the classification scheme.

Reviewer #3 (Remarks to the Author):

The authors report a new learning strategy named "Stochastic Emergent Storage (SES)" that uses a similarity-based criterion to select prototypes matching the target memory from random patterns generated by a photonic scattering medium. A weighted sum of the synaptic matrices corresponding to these prototypes is performed to construct an emergent archetype of the target memory. The proposed approach is innovative because it allows for storing information in a stochastic manner leveraging disordered scattering structures, while typical approaches store information in a deterministic manner. The manuscript is well-written, sufficient background information is provided, the methodology is scientifically sound, sufficient theoretical and experimental details are provided to allow for reproducibility and to support the conclusions. However, this work at its current state presents some limitations discussed as follows, preventing immediate use in practical information storage applications and restricting its impact. Therefore, it is advised that this work should be published in a well-established technical journal, rather than in the Nature Communications journal.

Major comments:

- The proposed strategy relies on typical computer-based procedures for computation and information storage in practise. A computer memory is needed to store the transmission matrices, limiting the function of the disordered scattering structures to a previously-reported classifier

function. Furthermore, it is still a software-based implementation rather than a hardware-based implementation. The authors mentioned that the coefficients of the linear combination of random optical synaptic matrices might be realized in a hardware fashion based on mode-specific neutral density filters. However, such implementation might result challenging in practise when considering large sets of matrices and increase the footprint of information storage devices. Overall, the proposed strategy has features of innovation, but its implementation in real-world information storage devices is still an unsolved issue.

- A discussion on the implications in terms of footprint and scalability of information storage devices based on the use of disordered scattering media is missing from the current version of the manuscript.

- At its current state, this work does not provide sufficient evidence about the advantages of the proposed approach compared to typical approaches for information storage applications (either magnetic-, electronics-, or optics-based approaches) in terms of key parameters such as storage capacity, throughput, and energy consumption toward application in information storage devices.

- The authors discuss the results presented in this work in relation to the formation processes of biological memory. However, as the authors mentioned, the challenge of understanding the biological memory formation process is a matter of ongoing debate. It is suggested to remove claims related to this aspect as they fall beyond the scope of this work. Also, the combined use of the terms "deterministic" and "random" to describe a phenomenon seems contradictory because determinism and randomness are typically seen as opposing concepts.

Minor comments:

- The terms "stochastic Emergent Storage" should be "Stochastic Emergent Storage".
- The terms "stochastic Hebb's storage" should be "Stochastic Hebb's Storage".
- The terms "optical synaptic matrix" should be used consistently throughout the manuscript.
- The acronyms "CAM" and "SES" should be used consistently throughout the manuscript.
- Page 4: the definition of the storage capacity of the proposed learning strategy should be clearly stated.
- Page 4, Figure 3C: the manuscript could be improved by providing a discussion and related references about strategies to improve the signal-to-noise ratio in the classification process of the proposed learning strategy.
- Page 5, the term "Figure 4f" should be "Figure 4e".
- Page 7, Supporting Information, the terms "modulation intensity" should be "modulation, intensity".

REVIEWER COMMENTS

Reviewer #1 (Remarks to the Author):

The author has utilized the naturally occurring random structure to store arbitrary patterns in this paper. In particular, in this work, the authors introduce the idea of stochastic emergent storage to overcome one of the challenges of the Hopefiled network, namely, the capacity limit in the densely populated network which rises from the emergence of uncontrolled memory states.

The SES utilizes a database for random patterns generated by a disorder and self-assembled structure such that a set of prototypes resembles the target memory via similarity criterion.

Next, the author also leverages unique properties of photonic computation such as wavefront transformation to implement the SES protocol experimentally without any additional fabrication.

More specifically, In this SES optical-based hardware, the optical transmission matrices allow one to perform both pattern storage and pattern retrieval in a way that an input can be compared to thousands of stored elements via a two-layer architect leading to deep classification.

To provide detail on the implementation, the author provides basic knowledge of wavefront scattering and discuss how one can reconstruct the optical synaptic matrix. Below I list few questions.

We extend our heartfelt gratitude to the Referee for their meticulous review of our manuscript and for the valuable suggestions they have provided to enhance the quality of our work. In the following sections, we will address each of the Referee's questions and comments in a point-by-point manner.

The main figures in the paper should be more comprehensive and explained in an easy way to convey the message directly without too much reference to the appendix or main text. In some sense, the figure should have enough captions that can be considered as complementary to the main text. I believe the author can improve the flow of figure 1 and its caption to make it more accessible. For example, it is not clear what is the scattering medium and what is the role of all different lenses.

The Referee suggests improving Figure 1's clarity.

First we have increased panel space and clarified the position of the scattering medium accordingly. These changes enhance the figure's visual clarity.

OLD FIGURE

NEW FIGURE

also extended and improved the caption including more details also about role of the lenses, and description of the symbol we employed:

FIG. 1. **Emergent memory storage scheme.** The sketch describes both the input query ϕ presentation and the measurement of the optical synaptic matrix V^ν (details in the Methods section). The first set of lenses (A-B) demagnifies (by a factor of 0.3) the DMD image, accommodating the scrambled input pattern of the scattering medium within Lens1's field of view. The second set of lenses (1-2) images the opaque medium's backplane onto the camera plane, with a magnification (factor of 11) that ensures the 1 speckle grain/mode per pixel imaging regime. Panels (a), (b), and (c) illustrate the process of instantiating a memory in our architecture. Panel (a) represent the similarity selection stage, wherein optical synaptic matrix (V^ν) are chosen based on their similarity with the target memory (ϕ^*). Panel (b) illustrates the construction of an emergent memory through the summation of relative optical synaptic matrices ($\sum^M V^\nu = J_{\phi^*}^{\mathcal{M},\lambda}$), resulting in the memory element ξ_Σ , obtained getting the largest eigenvector and computing the sign function (SEIG function). Panel (c) shows the pattern to be instantiated in memory ϕ^* , its vectorization, and the corresponding coupling matrix constructed using Hebb's Rule (as employed in SHS).

Also caption of figure 2 has been improved

FIG. 2. **Photonic Stochastic Hebb's Storage.** SHS realizes arbitrary optical operators emulating the target matrix T that stores the pattern ϕ^* (as exemplified in Fig 1c). Our experimental setup utilizes 9×9 binary patterns ($N = 81$, $M^L = 65536$) and M modes or camera pixels. Panel (a) presents the Mean Squared Difference (MSD) between the target and probe artificial optical synaptic matrices plotted against M/N . In panel (b), we show the Storage Error Probability versus M/N . Finally, in panel (c), we illustrate the Recognition Error Probability as a function of M/N . The insets within the figure display images of the reconstructed $J_T^{\mathcal{M},\lambda}$ matrix for various values of M . Note that the point at $M/N \sim 50$ is out of scale because of a very low error rate.

Also, the similarity selection is an important part of the work and I guess, needs to be clarified in the main text.

As suggested by the referee, we have incorporated a new paragraph into the main text specifically addressing the process of similarity selection

To perform the similarity selection with the optical modes the target pattern ϕ^* is compared with the eigenvectors of all the modes in the repository of characterized modes \mathcal{M}^L . The comparison is driven by the parameter S^ν

$$S^\nu = \hat{\phi}^* \cdot \hat{\xi}^\nu \quad (8)$$

that quantifies the degree of similarity between the first eigenvector of mode ν , ξ^ν , and ϕ^* .

The modes ν providing the higher S^ν are selected to feed a restricted repository of modes \mathcal{M}^* . The correspondent eigenvectors ξ^ν can be seen as prototypes of the target archetype, i.e. imperfect representations of the pattern to be stored (such as the one in Fig. 1c).

The main quantities on the figure are required to define in the main text and the figure also requires more detailed discussion.

In the caption of Figure 1, we introduced the description of all the symbolic quantities employed in the figure, and extended the description.

More specifically why the performance has different behavior in decreasing of storage vs classification, Naively, I would have thought the classification error for the same ratio of M/N should stay larger.

We have updated the terminology from 'Classification Error Probability' to 'Recognition Error Probability,' as the latter term more accurately conveys the meaning of the observable we have designed.

Here the referee is referring to Figure 2 and 3 where Storage Error Probability and the former Recognition Error Probability are reported for SHS and SES respectively.

These two quantities provide insight into two very different properties of our Storage Platform. Storage Error Probability (S_ERR), reports the number of errors (difference between the pattern to be stored and the pattern effectively stored) appearing in the storage process. In other words, S_ERR quantifies the number of features which assume a wrong value.

Recognition error probability (R_ERR), instead, is based on memory retrieval tests in which many patterns (from a repository) are proposed to a memory. Between the many patterns also lies the “right one” i.e. the one from which the memory is constructed. The pattern generating the higher value of the Transformed Intensity is the “recognized pattern”. R_ERR accounts for the ratio of error recognition upon multiple memory retrieval tests.

The S_ERR influences R_ERR : i.e. if many errors are present in the pattern injected in a repository the Recognition is more prone to failure. However R_ERR is also based on other features such as for example the nonlinear nature of the intensity (field modulus square) which tend to differentiate similar patterns and also the structure of the repository (if the repository contains very similar patterns then the classification is more difficult). Thus the relation is not a simple proportionality, while the two observables look at two very different aspects of the memory process i.e. storage fidelity and classification efficiency.

We added the following paragraph to clarify the point in the text

Note that Storage Error Probability (S_ERR) and Recognition error probability (R_ERR) provide insights on two very different aspects of our storage platform performance. S_ERR is essentially a storage fidelity observable, counting the ratio of wrong/correct pixels in the pattern to be stored which differ from the target memory to be stored ϕ^* , and accounts for the efficiency of our approach (the emergent storage) to instantiate a target memory in a memory repository. R_ERR retrieves recognition efficiency, thus reports on the ratio of memory retrieval tests providing wrong memory addresses, when different input patterns from a repository are proposed as stimuli. The S_ERR influences R_ERR : i.e. if many error are present in the pattern injected in a repository the recognition fails. However REP is also affected by other features such as for example the order of nonlinearity (we use intensity do appreciate differences in the field thus we employ a second order nonlinearity) which

influences the capability to differentiate similar patterns and also the structure of the repository (if the repository contains very similar patterns then the recognition task is more difficult). Thus the relation is not a simple proportionality, while the two observable look at two very different aspects of the memory process i.e. storage fidelity and recognition efficiency.

In Deep-SES instead, a single probe pattern is compared with many memories. We performed this task with digital data analysis but all the processes can be realized analogically, by performing pixel selection and weighting with DMDs. In such a case the probe pattern is directly tested *against* many memories: all the ones composing the *training set*. For the 9 class digit classification reported in the main paper Fig4, 3939 individual memories (441 per class) have been used. Employing a DMD with 33kHz frame rate would mean essentially performing optical classification in 0.1 seconds.

Maybe I am missing the point here, but would appreciate it if the author can elaborate on how in SES, one can impose the constructive interaction of the prototypes.

We concur with the referee's assessment, acknowledging the significance of this point. On page 3, we have incorporated an expanded explanation outlining the methodology behind optical synaptic matrix manipulation.

synaptic matrix. This method is based on the realization of a sensor collecting the *transformed intensity*

$$I^{\mathcal{M}}(\phi) = \sum_{\nu} \lambda^{\nu} I^{\nu}(\phi) \quad (4)$$

resulting from the incoherent sum of M intensities realized from that many transmitted optical modes from \mathcal{M} which is a subset of all the modes monitored \mathcal{M}^L . Coefficient λ^{ν} ($\in \{0 - 1\}$ and identified by a 4 bit positive real number) represent attenuation coefficients realized by mode-specific neutral density filters. Then employing the eq. 2 in eq. 4 we obtain

$$I^{\mathcal{M},\lambda}(\phi) = \phi \cdot \left(\sum_{\nu} \lambda^{\nu} V^{\nu} \right) \cdot \phi^{\dagger} = \phi \cdot J^{\mathcal{M},\lambda} \cdot \phi^{\dagger}. \quad (5)$$

Then we propose two techniques to design the optical operator $J^{\mathcal{M},\lambda}$: 1) the Stochastic Hebb's Storage (SHS) which enables to realize an arbitrary optical operator, 2) the Stochastic Emergent Storage (SES) which is instead aimed to the realization of optical memories.

Moreover, we further detailed in page 4 (the main text) how the memories interactions happens in SES:

In SES, these prototypes interact constructively, generating a representation of the memory ϕ^* in an emergent fashion [14]. The interaction is obtained by the incoherent sum of the intensity of several pixels/modes with proper attenuations/weights λ

In figure 3, again, would the author explain why the SES enhances the classification error a few order of magnitude better?

We have recognized the critical importance of this aspect of our paper, which was previously insufficiently clarified in the main text. Consequently, we have introduced a dedicated paragraph addressing the efficiency difference in our latest revision.

Thus, in summary, SHS enables to create an optical operator of arbitrary nature, which can effectively execute diverse tasks. This versatility arises from its capability to construct an artificial optical synaptic matrix designed by the user, effectively emulating a matrixial operator T . Conversely, SES focuses its functionality on generating an operator designed primarily for memory storage, excelling in this singular aspect. Consequently, it demands significantly less computational power and a smaller optical hardware setup (with a smaller M^* , see below).

This distinction influences the optimization procedure: SHS optimization relies on distances between matrices (measuring such distance computational cost scales as $N \times N$), while SES optimization is driven by distances between vectors (measuring such distance computational cost scales as N). Secondly, SES leverages preliminary similarity selection to identify the most relevant pixels/modes, a feature absent in SHS. As a result, the modes chosen for SES provide higher contrast in the classification task, especially in the $M < N$ regime. In contrast, in the $M > N$ regime (more degrees of freedom than constraints), both approaches achieve essentially the same level of efficiency.

Part c and d, require the information in the caption.

The part regarding panels c and d of figure 3 appears updated and highlighted in this updated version of the manuscript

FIG. 3. **Photonic Stochastic Emergent storage.** Top panels show the stored patterns obtained with SES for different values of M^* with Similarity selection (three patterns on the left), and with random selection (pattern on the right). Panel a) shows *Storage Error Probability* while panel b) the *Recognition error probability*. Both with respect to M^*/N . c) and d): Transformed intensity for 600 patterns in the repository (pattern index j on the ordinate axis) for the SES with the similarity selection (Sel.) stage and SES without the similarity selection. The mode $j = 241$, (with red circled intensity), correspond to the stored pattern. The insets between panels c) and d) report the obtained $J_{\phi^*}^{\mathcal{M}^*, \lambda}$

The author mentions that exploring the emergent learning process can provide insight into memory formation. This is indeed a very interesting point, maybe could have been a bit elaborated. In particular, the connection of SES with connectionist hypothesis.

We completely agree with the referee, thus we further enhanced the bridging with the memory formation hypothesis in the discussion:

The results presented in this study contribute to the ongoing challenge of understanding the biological memory formation process. **At the present time there are two major hypotheses that are the subject of debate**, the connectionist hypothesis [5], which suggests that neural networks form new links or adjust existing ones when storing new patterns, and the innate hypothesis [32], which posits that patterns are stored using pre-existing neural assemblies with fixed connectivity. **One central aspect in this ongoing debate pertains to the 'efficiency' of the network**, a facet that, in both artificial and natural networks, immediately invokes considerations related to energy consumption. On one side, it has long been established that in Hebbian networks, the number of memories (W) scales linearly with the number of nodes (N), expressed as $W = \alpha N$. For this reason, many research efforts are dedicated to optimizing the proportionality constant α . However, it appears that the latter has an upper limit of two. On the other side, it has been recently demonstrated, both numerically [7, 9] and theoretically [33, 34], that in the stochastic innate approach, the number of memory increases exponentially with the number of nodes: $W \propto e^{\alpha N}$. In other words, for larger system sizes, the innate approach predicts a significantly greater number of memories compared to the connectionist perspective. The "complexity" of the system (artificial neural network or brain), denoted as $S = \lim_{N \rightarrow \infty} \log(W)/N$, tends to zero for the connectionists, whereas it remains non-zero for the innatists.

SES introduces a fresh perspective to the problem by leveraging the Hebbian structure of the synaptic matrix, with a foundation of the connectionist hypothesis. However, SES goes beyond by exploring the potential of a stochastic *innate* network in which, **pre-existent** random synaptic structures are combined to generate memory elements in an emergent manner. **Whether the SES could bring a new point of view, lumping together the innatism and connectivism, is a fascinating hypothesis, that must be explored in the future.**

While I find the main results and effort of the authors very interesting and great potential publishing, however, I feel there is a need to improve the presentation and the flow of the work to make it accessible to readers, given the interdisciplinary nature of the work.

We have made substantial enhancements to the manuscript, addressing all of the referees' points and significantly improving its overall readability. As a result, we firmly believe that the paper is now well-prepared for full acceptance.

Reviewer #2 (Remarks to the Author):

The authors propose and demonstrate a new concept for optical deep classification exploiting the intrinsic disorder arising from a random diffuser scattering. They also pioneered a similarity-based algorithm to reduce the number of basis elements and, thus, reduce the computation. This is an interesting approach that gets rid of the complexity impose on the diffractive elements in similar free-space optical approaches, and instead moves it to conventional computer-based post processing tasks. This could translate into a simplification of the experimental setup to perform this type of tasks.

I am in principle supportive of this manuscript to be accepted for publication in Nature Communications

We express our gratitude to the referee for their overwhelmingly positive review.

Furthermore, we have recognized the pivotal role of classification in our research and have taken additional measures to enhance the characterization of our disordered classifier, including an extended comparison with relevant competitor architectures (details provided below in our point-by-point reply).

It is essential to emphasize that the results presented in this paper extend beyond the realm of classification and analog similarity retrieval based on intensity. Indeed our system offers a range of versatile capabilities:

- It introduces a novel approach for reading and writing user-designed optical memory patterns within a disordered medium.
- By replicating the writing process, it becomes feasible to create a substantial number of memories, effectively transforming a disordered medium into a Content Addressable Memory.
- It enables a novel training strategy that simplifies the process by requiring the grouping of memories belonging to the same class.
- It enables lossy information compression for the installed photonic memories (See new section in supplementary “Data compression with SES”)

These aspects highlight the broader implications and potential applications of our research beyond the specific classification paradigm.

However, I have many questions for which I couldn't find answers, several details could use more in detail discussion (e.g., the limitations of the approach, the difference between SHS and SES, etc.), and there is no benchmarking with respect to similar approaches so a direct comparison renders difficult. I suggest a revision of the structure and discussion of the manuscript to address the following points. In particular, I suggest the authors use subtitles to clearly separate the different sections and have clearer transitions, as found below, I'm particularly confused by Fig. 2 and Fig. 3 and the actual differences between SHS and SES.

We have taken significant steps to enhance the readability of our paper. These improvements include a division of the text into section and subsection, the incorporation of a figure of merit, and the addition of a comparison table between SES and two competing architectures.

To provide a more comprehensive understanding of our research, we have also refined the description of the differences between SHS and SES. Moreover, we have clarified the distinction between SES technology, which is capable of storing a single memory within the scattering medium, and Deep-SES, which can employ stored memories to generate classes by grouping them.

For a detailed account of these modifications, please refer to our individual responses to the referees' questions below.

• The number of measurements and computations needed in this approach come with a penalty in processing time, although this, which is basically “training,” is time consuming regardless of the platform. How does this approach compare to others? Suggesting a figure of merit and comparing to other approaches can be useful to understand the real advantage of the method.

We greatly appreciate the referee's insightful comment, which has drawn our attention to a crucial aspect of SES and Deep-SES that we had overlooked in the previous version of the paper. Indeed, as the referee rightly points out, by conducting a preliminary pixel/mode selection, we can significantly enhance the efficiency achieved for a given number of camera pixels.

To illustrate this improvement, we have introduced a new figure of merit as a panel in Figure 4 (panel 4f). This panel reports the classification efficiency versus the number of employed camera pixels/Modes, denoted as M^* . The significance of M^* extends to the classification protocol's complexity, impacting both the hardware resource cost and the computational demands of the training process.

As an example, to achieve a classification efficiency of over 90% for digit recognition, our Deep-SES architecture requires $M^* = 40$ camera pixels. In contrast, a competitor architecture that performs classification with light speckles (such as RRS, as mentioned in the paper) demands a substantially larger number of pixels, with $M^* = 1600$ needed to attain the same result. This remarkable disparity underscores the efficiency and practicality of our approach. The new panel is reported also below

and the updated the caption of Fig 4

Panel f) reports the classification efficiency on the same digit database for Deep-SES and the Ridge Regression with Speckles (RRS) [21], versus the number M^* (see Methods).

The main text has been also updated (page 6):

shows integrated intensity after threshold. Figure 4e reports the confusion matrix for all labels, demonstrating categorical recognition efficiency higher than 90% which eventually may be enhanced employing error correction algorithms [31]. This result demonstrates the possibility to generate deeper optical machine learning architectures and perform training by simply grouping memories. The potential of Deep-SES is further demonstrated by panel 4f, where we report a figure of merit comparing the efficiency of Deep-SES with Ridge regression with speckles (RRS) [21] (simulated). Note, while Deep-SES reaches an efficiency 90% for $M^* = 40$, the RRS surpasses this threshold for $M^* = 1600$. As M^* represent the number of physical camera pixel employed in the classification, SES is capable of delivering a classifier with a much smaller hardware and computational complexity. The origin of this advantage emerges from the fact that our memory writing process, selects pixels/modes which are the most correlated with the pattern to be recognized thus outperform with respect to randomly chosen ones. Moreover deep-SES enables thus to reorganize memories into new classes (reshuffling of classes) with almost no computational cost, a task which typically requires a new training in standard digital or optical architectures (see **methods**).

A new Section has been added to the supplementary information file “Comparison with other optical platforms” which comprehends a table (Table 1) SES and Dep-SES with RRS and also with the DDNN platform:

Comparison with other optical platforms

We have conducted a comprehensive comparison of our results with other architectures that rely on free-space transformations, specifically Diffractive Deep Neural Networks (D²NN) [1] and Ridge Regression With Speckles (RRS) [2].

D²NN, as reported in [1], is based on the design of transmission matrices achieved through iterative heavy-weight computation of optical propagation. This process involves multiple optical simulations, and the resulting node weights are subsequently translated into physically printed scattering layers for optical experiments. Notably, the creation of the artificial disordered structure in D²NN demands substantial computational effort and a subsequent fabrication stage, both of which are unnecessary in both SES/Deep-SES and RRS.

RRS instead requires strong computational effort (inversion of a big size matrix) and an high number of de-

tectors (large M^*).

In Table I, the second column outlines the capability to read and write optical memories. With SES, we have demonstrated the capability for both writing and reading (W/R) memories, along with the ability to store up to 4096 memory elements without any fabrication effort. In contrast, RRS does not possess memory storage capabilities. Although DDNN inherently provides the possibility to write hardware memories (even though this aspect may not be explicitly detailed in the paper), it’s worth noting that there is an intrinsic advantage in having a memory instantiated within the optical layers, as these layers are readily amenable to further optical operations.

The third column of Table I outlines the classification performance for $M^* = 120$. It’s important to note that in both SES and RRS, M^* represents the number of camera pixels employed, thus directly influencing the hardware resource requirements and overall protocol complexity. Additionally, it’s worth highlighting that DDNN operates on a different paradigm, where M^* does not serve as the relevant parameter for evaluating computational-experimental complexity.

The fourth column of Table I reports on the ability to obtain analog output. Both DDNN and SES produce outputs in the form of optical intensity, making them compatible with subsequent analog operations, potentially enabling faster processing. The paper on RRS [2] does not provide information on this capability.

In SES, the ‘transformed intensity’ corresponding to each memory serves as an analogical measure of the probe’s degree of similarity with the stored memory. This unique feature enables the realization of Content Addressable Memory (CAM), which not only provides the address of the most relevant retrieved memory but also quantifies the degree of similarity with other stored memories. It’s important to note that this special capability is unattainable in the other two platforms, as documented in the fifth column of Table I.

TABLE I. Comparison Between SES, Deep-SES, RRS[2], DDNN[1]. Second column reports about the Write capability(W) and Read capability (R). The third column report efficiency is (defined as the ratio of the number of elements appearing in the diagonal in the confusion matrix with respect to all the elements of the same matrix). Note that for DDNN (efficiency appearing with an asterisk in the third column), the dataset is different that used in SES, Deep-SES, and RRS. “Analog output” column report the possibility to gather the output in the form of intensity. The third column report the possibility of the platform to work as Content Addressable Memory, and to employ Intensity as a measure of the similarity between the stored memory and the probe pattern.

Architecture	Memory Storage	Efficiency ($M^* = 120$)	Analog output	CAM and Similarity
SES	W/R	-	✓	✓
Deep SES	W/R	91.71 %	✓	✓
RRS[2]	none	53.54 %	✗	✗
DDNN[1]	W	91.75* %	✓	✗

• What is the purpose of the 0.3 x de-magnifications? The image of each DMD pixel after demagnification is about half the mean free-path of the diffuser. Is there an intrinsic limitation for the method in terms of how these two parameters compare? That clearly determines the scattering, but how does that affect the technique and its precision?

We thank the referee for addressing this important point. We have made significant improvements to the description of our experimental design, providing a more

comprehensive understanding of the rationale behind our choices in magnification, scattering medium thickness, and camera Region of Interest (ROI). This is the revised statement

one-mode-per-pixel configuration (one-pixel-per-speckle-grain). The optical collection apparatus, does not require a particular performance, indeed we employed a commercial, low-cost 25.45 mm focal bi-convex lens for the light collection from the far side of the sample. Several constraints have to be considered in the experimental design. When light from a DMD super-pixel emerges from the disordered medium, it is diffused into a larger disk-shaped area. For this reason, we have to ensure that each these light disks are interfering with all the disks generated by other super-pixels in the detection camera ROI, and this introduces a constraint on the maximum ROI size (M^L). The size of these diffusion disks is regulated by the thickness of the disordered scattering medium. Nevertheless, note that increasing the scattered thickness also decreases the light intensity on the camera and the stability of the speckle pattern thus a trade-off between thickness and signal-stability should be found at the experimental design step.

• Continuing with the magnification, the second magnifying system is design to achieve 11x. The authors mention that this is to “minimize the speckle size”. Wouldn’t it maximize it if we think in terms of size (in a way that it fits entirely within the detector size) or what do the authors mean with minimize?

We thank the referee for addressing this point. In our statement, we were referring to the size of the speckle grain rather than the entire speckle pattern. As the referee correctly highlights, the size of the speckle grain plays a significant role in determining the size of the entire random memory repository, denoted as M^L . Additionally, it's essential to ensure that there is one speckle grain per pixel, as this minimizes the correlation between neighboring pixels, modes, and random memories.

We amended the text:

detection camera ROI. Then, the back layer of the disordered structure is imaged on the detection camera ($11 \times$ magnification). This magnification has been chosen to minimize the speckle grain size in order to work in the one-mode-per-pixel configuration (one-pixel-per-speckle-grain). The optical collection apparatus, does not require

• I am confused about the selection of the random subset of (cursive) M out of the full M^L in Fig. 2 for SHS. Why was it introduced in first place? Is the random subset vs. similarity selection the only difference between SHS and SES? I might have missed a big point here.

The referee highlights the need for further clarification regarding the distinction between SES and SHS.

Indeed, the primary differentiator between these two techniques lies in their intended tasks. Specifically: SHS possesses the capability to create an optical operator of arbitrary nature, which can effectively execute diverse tasks. This versatility arises from its capacity to construct an optical synaptic matrix designed by the user, effectively emulating a matricial operator T .

Conversely, SES focuses its functionality on generating an operator designed primarily for memory storage. In essence, it excels in this singular aspect. Consequently: SES demands significantly less computational power and a smaller optical hardware setup (a smaller M^*).

However, it is tailored exclusively to the purpose of storing a single memory pattern.

This distinction influences the optimization procedure: SHS optimization relies on distances between matrices ($N \times N$), while SES optimization is driven by distances between vectors (N). Secondly, SES leverages preliminary similarity selection to identify the most relevant pixels/modes, a feature absent in SHS. As a result, the modes chosen for SES provide higher contrast in the classification task, especially in the $M < N$ regime. In contrast, in the $M > N$ regime (more degrees of freedom than constraints), both approaches achieve essentially the same level of efficiency.

To clarify this aspect we introduced the following sentence in

Thus, in summary, SHS enables to create an optical operator of arbitrary nature, which can effectively execute diverse tasks. This versatility arises from its capability to construct an artificial optical synaptic matrix designed by the user, effectively emulating a matricial operator T . Conversely, SES focuses its functionality on generating an operator designed primarily for memory storage, excelling in this singular aspect. Consequently, it demands significantly less computational power and a smaller optical hardware setup (with a smaller M^* , see below).

This distinction influences the optimization procedure: SHS optimization relies on distances between matrices (measuring such distance computational cost scales as $N \times N$), while SES optimization is driven by distances between vectors (measuring such distance computational cost scales as N). Secondly, SES leverages preliminary similarity selection to identify the most relevant pixels/modes, a feature absent in SHS. As a result, the modes chosen for SES provide higher contrast in the classification task, especially in the $M < N$ regime. In contrast, in the $M > N$ regime (more degrees of freedom than constraints), both approaches achieve essentially the same level of efficiency.

• Fig. 2: we know $N=81$, but what is N ? N is somehow defined in the methods, but at this point in the main text the reader doesn't know what N is. Moreover, in the methods, the authors suggest $T=N \times N$ but later on, $N \times N$ is also the number of superpixels in the DMD, is this correct?.

• Fig. 2: M is the number of random samples, isn't it?. The meaning of M and N should be added to the caption.

N and M are important parameters for both SES and SHS. This is the number of features (pixels) in the pattern. To present a pattern to our disordered classifier, we need to prepare N

superpixels onto the DMD. M^L is the total number of modes measured, thus is identical to the number of active pixels on the camera. We clarified in the main text and in the figure 2 caption the meaning of both N and M :

The idea stems from the fact that intensity scattered by a disordered medium into a mode ν resulting from an input pattern $\phi \in \{0,1\}^N$ (with N size of the input pattern) may be written as:

$$I^\nu(\phi) = \phi \cdot V^\nu \cdot \phi^\dagger \quad (2)$$

with the scattering process driven by the matrix $V^\nu \in \mathbb{C}^{N \times N}$;

$$V^\nu \sim \xi^\nu \otimes \xi^{\nu\dagger} \quad (3)$$

generated from the tensorial product of the transmission matrix row (transmission vector) $\xi^\nu \in \mathbb{C}^N$ with its conjugate transpose $\xi^{\nu\dagger}$.

resulting from the incoherent sum of intensities from the subset of modes \mathcal{M} composed by M elements. \mathcal{M} is a randomly selected subset of all the modes monitored \mathcal{M}^L .

FIG. 2. Photonic Stochastic Hebb's Storage. SHS realizes arbitrary optical operators emulating the target matrix T that stores the pattern ϕ^* (as exemplified in Fig 1c). Our experimental setup utilizes 9×9 binary patterns ($N = 81$, $M^L = 65536$) and M modes or camera pixels. Panel (a) presents the Mean Squared Difference (MSD) between the target and probe artificial optical synaptic matrices plotted against M/N . In panel (b), we show the Storage Error Probability versus M/N . Finally, in panel (c), we illustrate the Recognition Error Probability as a function of M/N . The insets within the figure display images of the reconstructed $J_T^{\mathcal{M},\lambda}$ matrix for various values of M . Note that the point at $M/N \sim 50$ is out of scale because of a very low error rate.

• Fig. 2: there is not discussion of the meaning of these results in the main text. The authors simply show it but do not discuss what it means, for example, to achieve low error for $M > 10N$ or why $M=N$ (which is a significantly small number for M) yields lower classification error, like they did in Fig. 3. They also mention that SES is basis hungry, but do not elaborate on this statement. If $M=10*N$ with $N=81$ is enough to have low errors, then the basis is not that big after all if we consider that M is initially over 65k matrices.

As requested by the referee we improved the explanation of the Hebb's limits, and accounted for the lacking point in the logarithmic graph (negligible error is not reported).

ory element. This is connected to the fact that the target matrix T is constructed on $N \times N/2$ parameters (is symmetrical) acting as constraints, while we have M free parameters to emulate it. A full emulation of of T is expected thus to be successful for $M > N \times N/2$ which is consistent with what we retrieve in Figure 2 (Data for $M=4096$ are not reported because error is negligible both for storage and classification.)

• Fig. 2: the inset in Fig. 2b shows $M=4096$, however the data point $M/N \sim 50$ is not plotted. The max is 10. Why?

Note that the scale is Y-logarithmic so very small errors (or no errors) are appearing out of scale in the plot. see also the text just above.

• If I am understanding correctly from Eq. (5) and Eq. (6), the results in Fig. 2 for random subsets of M with $N=81$ should be equivalent to SES for M^* random subsets. However, Fig. 3 shows, for the same N and a random basis, much worse results than in Fig. 2. What is different and why?

The referee poses a question about the consistency between results reported in figure 2 and results in figure 3 relative to SES with random basis (with similarity selection replaced by random decimation).

The two aspects are strongly different because the gradient descent process which is in charge of realizing λ^ν weights works in different ways, i.e. minimizing a different cost function to get very different targets. In SHS the cost function is driven by the MSE between the target matrix T and the Probe matrix J_T . In SES we just compute the distance between the first eigenvector of J_{ϕ^*} and the target memory to be installed ϕ^* . Being these two completely different tasks, it is understandable how the two ancillary observables (the storage error and the classification error) provide different results.

To further clarify the difference between SHS and SES, we improved figure 2 adding an additional panel reporting the final MSE value obtained after optimization versus M and reorganizing the insets.

FIG. 2. Photonic Stochastic Hebb’s Storage. SHS realizes arbitrary optical operators emulating the target matrix \mathbf{T} that stores the pattern ϕ^* (as exemplified in Fig 1c). Our experimental setup utilizes 9×9 binary patterns ($N = 81$, $M^L = 65536$) and M modes or camera pixels. Panel (a) presents the Mean Squared Difference (MSD) between the target and probe artificial optical synaptic matrices plotted against M/N . In panel (b), we show the Storage Error Probability versus M/N . Finally, in panel (c), we illustrate the Recognition Error Probability as a function of M/N . The insets within the figure display images of the reconstructed $\mathbf{J}_T^{\mathcal{M},\lambda}$ matrix for various values of M . Note that the point at $M/N \sim 50$ is out of scale because of a very low error rate.

Furthermore, we added in the supplementary material a section that discusses a geometrical interpretation that tries to show how SHS and SES are different. In this section, we show how SHS can be understood in terms of an orthonormal spase constructed on the pattern that we want to store phi and the independent basis added by the selected modes.

Geometrical Interpretation

In SHS and in SES, given a pattern ϕ^* and its respective synaptic matrix \mathbf{T} we want to derive a weighted sum of M matrices \mathbf{V}^ν that optimizes a certain cost function. In SHS the cost function is given by Eq. (29) and in SES it is given by Eq. (37). From the observation that $\mathbf{V}^\nu = \xi^\nu \otimes \xi^{\nu\dagger} + \eta^\nu \otimes \eta^{\nu\dagger}$, we know that each \mathbf{V}^ν has only two non-negligible eigenvalues-eigenvectors couples. If $M < N$, and if we assume that ϕ^* , ξ^ν , and η^ν are statistically independent N -dimensional vectors, we can assume that the collection of these vectors forms a $2M + 1$ di-

dimensional basis, $Q = \{\phi^*, \xi^0, \xi^1, \dots, \xi^M, \eta^0, \eta^1, \dots, \eta^M\}$. If we use Gram–Schmidt process we can derive an orthonormal basis P corresponding to Q , such that $P = \{\phi^*, v^0, v^1, \dots, v^K\}$, where $K = \min(2 * M, N - 1)$. It follows that we can write the synaptic matrix \mathbf{T} as:

$$\mathbf{T} = P\Lambda_{\phi^*}P^T \quad (1)$$

where

$$\Lambda_{\phi^*} = \begin{bmatrix} 1 & 0 & \dots & 0 \\ 0 & 0 & \dots & 0 \\ \vdots & \vdots & \ddots & \vdots \\ 0 & 0 & \dots & 0 \end{bmatrix}$$

Similarly, we can write

$$\begin{aligned} \mathbf{J}_{\phi^*}^{\mathcal{M}, \lambda} &= \sum_{\nu}^M \lambda_{\nu} V^{\nu} = \sum_{\nu}^M \lambda_{\nu} (\xi^{\nu} \otimes \xi^{\nu\dagger} + \eta^{\nu} \otimes \eta^{\nu\dagger}) \\ &= \sum_{\nu}^M \lambda_{\nu} P A^{\nu} P^T \end{aligned} \quad (2)$$

where A^{ν} is a $N \times N$ matrix

$$A = P^T (\xi^{\nu} \otimes \xi^{\nu\dagger} + \eta^{\nu} \otimes \eta^{\nu\dagger}) P$$

Consequently, in SHS, optimizing Eq. (29) is equivalent to optimizing $\mathcal{F}(\mathcal{M}, \lambda) = \sum (\Lambda_{\phi^*} - \sum_{\nu} \lambda_{\nu} A^{\nu})^2$. Clearly, as M tends to N we get to a perfect reconstruction of \mathbf{T} . In SES, the interpretation is a little less straightforward because we are optimizing the projection of ϕ^* with the largest eigenvector of $\mathbf{J}_{\phi^*}^{\mathcal{M}^*, \lambda}$. What happens is that we

tend to constructively sum the weighted projections of V^{ν} parallel to ϕ . Meanwhile, we can assume that the perpendicular terms of V^{ν} are randomly weighted. Thus the contribution of the perpendicular terms of V^{ν} vanishes for M growing to infinite, but it is not negligible for finite intermediate values of M . In conclusion, while in SHS we optimize simultaneously the parallel and perpendicular contribution of V^{ν} , in SES we optimize only the parallel contribution of V^{ν} . This is the reason why in Figure 2 and Figure 3, we see that given a random subset \mathcal{M} with M modes SHS performs better than SES. Nevertheless, the contribution of the perpendicular terms of V^{ν} is strongly reduced when we select the V^{ν} with the strongest values of similitude degree $\mathcal{S}^{\nu} = \hat{\phi}^* \cdot \hat{\xi}^{\nu}$, because we are selecting the V^{ν} with the smaller perpendicular component. This is why in Figure 3 we see a clear difference between using the random \mathcal{M} or the sorted \mathcal{M}^* .

- **What are the limitations of the approach in terms of spatial frequencies?**

The system does not impose specific criteria for optical resolution or lens numerical apertures since the presence of speckles smaller than a single camera pixel does not confer any notable advantages. The collection lens does not require exceptional spatial resolution performance. The following statement has been included in the Methods section to provide clarification on this matter.

speckle-grain). The optical apparatus for collection does not require a particular performance, indeed we employed a commercial, low cost 25.45 mm focal bi-convex lens for the light collection from the far side of the sample. Sev-

- **The process to perform the classification is not fully clear to me. How fast is the classification? What are the experimental and post processing components? I would appreciate a similar figure like Fig. 1 for the storage scheme, but with the classification scheme.**

As requested by the referee, we added an explanatory sketch, and a dedicated chapter in the methods, including all the requested information about, pattern recognition, classification and timing. We also improved the wording of our observables. In the previous version of the paper we designed a quantity named "Classification error probability", (employed in fig 2 and 3) this has been reworded as "Recognition error probability". This is because the "recognition" is more similar to the task effectively performed to measure the quantity, and also to avoid confusion with the true classification process which is instead reported in figure 4.

FIG. 5. Sketch of the pattern recognition and classification. The task is to assess which pattern into a set of probe patterns corresponds to one specific pattern memorized into the disordered classifier. This stage is carried on after the memory writing process has already been finalized, then the set of most relevant pixel/modes and the specific weights lambdas are already known. Then the first probe pattern is presented to the disordered classifier through the DMD. Intensity from the selected M^* pixels is retrieved and summed with digital operation or analogically. The retrieved transformed intensity, provides an analogical measure of the degree of similitude between the probe and the stored pattern ϕ^* . This process is then repeated for all the patterns in the set of of probe patterns. The recognized pattern is the pattern producing the higher intensity.

Note that Storage Error Probability (S_ERR) and Recognition error probability (R_ERR) provide insights on two very different aspects of our storage platform performance. S_ERR is essentially a storage fidelity observable, counting the ratio of wrong/correct pixels in the pattern to be stored which differ from the target memory to be stored ϕ^* , and accounts for the efficiency of our approach (the emergent storage) to instantiate a target memory in a memory repository. R_ERR retrieves recognition efficiency, thus reports on the ratio of memory retrieval tests providing wrong memory addresses, when different input patterns from a repository are proposed as stimuli. The S_ERR influences R_ERR : i.e. if many error are present in the pattern injected in a repository the recognition fails. However REP is also affected by other features such as for example the order of nonlinearity (we use intensity do appreciate differences in the field thus we employ a second order nonlinearity) which

influences the capability to differentiate similar patterns and also the structure of the repository (if the repository contains very similar patterns then the recognition task is more difficult). Thus the relation is not a simple proportionality, while the two observable look at two very different aspects of the memory process i.e. storage fidelity and recognition efficiency.

In Deep-SES instead, a single probe pattern is compared with many memories. We performed this task with digital data analysis but all the processes can be realized analogically, by performing pixel selection and weighting with DMDs. In such a case the probe pattern is directly tested *against* many memories: all the ones composing the *training set*. For the 9 class digit classification reported in the main paper Fig4, 3939 individual memories (441 per class) have been used. Employing a DMD with 33kHz frame rate would mean essentially performing optical classification in 0.1 seconds.

Reviewer #3 (Remarks to the Author):

The authors report a new learning strategy named “Stochastic Emergent Storage (SES)” that uses a similarity-based criterion to select prototypes matching the target memory from random patterns generated by a photonic scattering medium. A weighted sum of the synaptic matrices corresponding to these prototypes is performed to construct an emergent archetype of the target memory. The proposed approach is innovative because it allows for storing information in a stochastic manner leveraging disordered scattering structures, while typical approaches store information in a deterministic manner. The manuscript is well-written, sufficient background information is provided, the methodology is scientifically sound, sufficient theoretical and experimental details are provided to allow for reproducibility and to support the conclusions.

However, this work at its current state presents some limitations discussed as follows, preventing immediate use in practical information storage applications and restricting its impact. Therefore, it is advised that this work should be published in a well-established technical journal, rather than in the Nature Communications journal.

We express our gratitude to the referee for the evaluation of our work and for acknowledging the positive aspects of our manuscript. With the improvements incorporated into this enhanced version, we firmly believe that our manuscript qualifies for consideration for publication.

Major comments:

- The proposed strategy relies on typical computer-based procedures for computation and information storage in practice. A computer memory is needed to store the transmission matrices, limiting the function of the disordered scattering structures to a previously-reported classifier function. Furthermore, it is still a software-based implementation rather than a hardware-based implementation.

The referee raises questions regarding the system's storage nature and performance in terms of digital memory resources. Notably, optical storage presents an inherent advantage due to its readiness for lightning-speed optical operations. Additionally, SES offers a memory advantage through lossy compression as described below.

However, it's important to clarify a statement made by the referee: The referee states that “A computer memory is needed to store the transmission matrices, limiting the function of the disordered scattering structures”. This statement is not entirely accurate.

The complete information about transmission matrix rows ξ^ν and optical synaptic matrices V^ν are indeed required just during the modes similarity selection and gradient descent processes for memory extraction. However, this information is not needed anymore after the writing process and effective storage of the memory itself requires much limited resources.

Indeed, what's essential to storage are solely the addresses (indices) of the pixels/modes that compose the M^* set and their corresponding weights λ^ν . To illustrate, consider a pattern with 256 bits. It can be stored using only M^* modes from a repository of $M^*L=65536$ modes, requiring 16 bits to store the mode address and 4 bits for the weight value. Consequently, the cost of storing a pattern in this scenario amounts to $20M^*$ bits, resulting in a compression advantage of up to $M^*=12$ (i.e. in the regime $20M^* < N$).

The performance of this lossy compression is detailed in a new section, along with the corresponding figure, as reported in the Methods.

Data compression with SES

Storing a memory with SES has a twofold advantage.

First, being the memory written in an optical layer, it is ready for further operations which may be performed at the speed of light, such for example classification via deep-SES. Second SES written memories provide a form of (lossy) data compression. To retrieve a memory we require the address of the pixels composing M^* selected modes (16 bits for a $M^L = 65536$ repository), and the values of the lambdas (4 bits) for the attenuation. This is a total of $20M^*$ bits per memory. For a pattern requiring N bits to be stored, this means in the regime $20M^* < N$ we obtain an effective data compression. Obviously this data compression comes at a cost, because SES storage comes with a certain error percentage (see figure 2a in the main paper). In the following Fig. 2, we report the compression performance versus M^* for an $N = 256$ pattern. In the green shaded area, there is a $1/(\text{compression ratio})$ smaller than unity thus highlighting the efficient compression regime.

The authors mentioned that the coefficients of the linear combination of random optical synaptic matrices might be realized in a hardware fashion based on mode-specific neutral density filters. However, such implementation might result in a challenge in practise when considering large sets of matrices and increase the footprint of information storage devices. Overall, the proposed strategy has features of innovation, but its implementation in real-world information storage devices is still an unsolved issue.

The referee's valid concern about the challenges associated with physically realizing the mode selection and mode-specific neutral density filters. We propose a method that can simultaneously overcome these difficulties by leveraging a second Digital Micromirror Device (DMD) within the modes plane, coupled with a relay system.

In this configuration, the second DMD serves a dual role, functioning both as a mode selector and as a generator of neutral density filters. It accomplishes this by utilizing large

superpixels (4x4) that can be incrementally activated to achieve varying degrees of attenuation.

As outlined in the paper (supplementary materials), our experimental results have been validated using mode-specific neutral filters with 16 different attenuation values, evenly spaced along a linear scale. These attenuation levels can be effectively implemented using the 4x4 superpixels on the second DMD, positioned in the mode space. Subsequently, the mode space, after undergoing spatial filtering, is re-imaged onto a camera sensor or a single, wide-area sensor.

This innovative approach not only eliminates the complexities associated with precise alignment procedures (alignment can be carried out after the hardware assembly by associating DMD pixels with mode indices) but also offers the advantage of rapidly probing different memories without the need for the intricate fabrication of distinct filtering masks or mask substitutions.

We are dedicated to exploring and studying this fully hardware-based version of SES and Deep SES in detail in the future. All pertinent details and results are comprehensively documented in a dedicated new chapter in the Methods section, titled "Full-Hardware SES and Scalability," accompanied by a figure with the setup scheme.

Full-Hardware SES and Scalability

SES can be implemented in two distinct versions: partially hardware-based and fully hardware-based. In the partially hardware-based version, as demonstrated earlier, the processes of pixel/mode selection and neutral density filtering to achieve the λ^{ν} coefficients are achieved through software-based multiplication, wherein individual intensity values I^{ν} are modulated by relative absorption coefficients.

In the fully hardware-based version, absorption coefficients can be realized using hardware masks that either fully absorb light for irrelevant modes or attenuate light by a factor of λ^{ν} through neutral density filtering for selected M^* modes. However, it's important to note that the fabrication of such masks can be challenging due to the requirement for micron-level precision during fabrication and fine alignment during assembly. These obstacles can be effectively overcome through the implementation of an adaptive optics layer.

This alternative approach, relies on a second Digital Micromirror Device (DMD), serving as a programmable and reconfigurable absorptive mask, referred to as a "mode selector" DMD. With this approach, a single optical setup can be employed to store multiple memories, eliminating the need for the individual fabrication of attenuation masks for each memory. When testing or probing against a memory, it becomes a matter of rearranging the mode selector DMD, rather than creating an entirely new optical setup with mode-specific absorptive masks. This streamlined approach enhances the flexibility and scalability of the SES system, making it more adaptable to various memory storage and retrieval tasks.

To achieve the required 4 bit depth required in the $\lambda^{\nu}u$ values, the mode selection DMD is organized into 4×4 superpixel, (so that each superpixel corresponds to a mode ν) which are turned on with a specific ratio of ON/OFF pixels to realize the required absorption. Eventually a second iris can be added after the mode selection DMD, in order to blur together, the grainy structure. This DMD based approach, not only enables a fabrication free approach, but also avoids the need for fine alignment, as the alignment can be performed, *ex-post* by mapping DMD superpixels on measured modes. Also

tical complexity. A sketch of the experimental scheme is provided in the Fig. 1 below.

this DMD-based approach, enables higher flexibility, so that a single probe pattern, can be "tested" against different memories at kHz speed.

FIG. 1. Sketch for re-configurable, and scalable SES. Optical Elements acronyms : IL: imaging Lens; BS: Beam Splitter; RM: Removable Mirror; M: Mirror. The probe pattern ϕ is injected in the disordered classifier by the probe DMD. Light at the back focal plane of the disordered medium is imaged on a Mode selection DMD, which transmits selected modes and introduces the weights $\lambda^{\nu}u$. A camera is present for the optical synaptic matrix, Transmission matrix measurement, while at the classification, memory retrieval stage the RM mirror is removed and the *transformed intensity* is retrieved by a single large area detector to increase the measurement speed. In the "Pattern probing" configuration, a single memory is "printed" on the Mode selection DMD while multiple probe pattern are cycled and *transformed intensity* is retrieved for each pixel thus retrieving the degree of similarity of a single memory against many probe patterns and retrieving the most pattern most similar to the chosen memory. On the contrary a single probe pattern can be tested against many memories, by cycling on the Mode Selection DMD (like in Fig 4 a.). Note that the the scalable Module part can be replicated enabling to test a single pattern simultaneously against many different memories in parallel.

By employing a commercially available DMD with 4 million pixels, 250 thousand modes can be easily probed at 10 kHz speed. This approach is also scalable: our experiment has been realized with a 0.5 Watts CW power laser while we tested transmission matrix stability up to 20 Watts illumination. Thus considering incremental losses, the scheme can be replicated on at least 10 times without any fundamental obstacle other than opt-

- A discussion on the implications in terms of footprint and scalability of information storage devices based on the use of disordered scattering media is missing from the current version of the manuscript.

We have included extensive details regarding the scalability of the proposed hardware in the newly added "Full-Hardware SES and Scalability" section, as mentioned earlier. The method enables the replication of transmitted light modes on various flexible DMD masks, facilitating simultaneous comparisons of a single probe pattern with multiple memories. Tests assessing the stability of the transmission matrix, indicate that our approach can be replicated up to 10

times when employing a high-power laser without appreciable decrease of the transmission matrix stability.

It is important to note that the number of modes that can be concurrently employed by a single device is constrained by the maximum size or number of pixels supported by the current DMD devices, which is typically around 4 million pixels. This scalability insight highlights the potential for parallel processing and the efficient use of resources in SES-based memory systems.

All these information are reported in the newly added section.

- At its current state, this work does not provide sufficient evidence about the advantages of the proposed approach compared to typical approaches for information storage applications (either magnetic-, electronics-, or optics-based approaches) in terms of key parameters such as storage capacity, throughput, and energy consumption toward application in information storage devices.

The referee requests to provide information about the distinctive feature of SES in contrast to other approaches.

SES operates on an emergent process for memory realization, eliminating the need for physical fabrication. In SES, the sole material requirement for memory writing is a disordered, self-assembled strongly scattering structure. To our knowledge, SES stands as the only optical memory writing and reading process that doesn't rely on direct optical fabrication or modification of properties of a physical structure.

The software version of SES is fully fabrication-less. The original ideas of hardware-SES requires instead a fabricated-mode specific neutral density filter mask, which thus introduces a form of fabrication. In the present version of the paper however, we introduce the new section **"Full Hardware SES and scalability"** in which the fabricated, mode specific, neutral density filter masks are replaced by programmable reconfigurable DMDs thus recovering the fully fabrication-less characteristic also for the hardware versions of SES. This process is described in the newly added section :

This alternative approach, relies on a second Digital Micromirror Device (DMD), serving as a programmable and reconfigurable absorptive mask, referred to as a "mode selector" DMD. With this approach, a single optical setup can be employed to store multiple memories, eliminating the need for the individual fabrication of attenuation masks for each memory. When testing or probing against a memory, it becomes a matter of rearranging the mode selector DMD, rather than creating an entirely new optical setup with mode-specific absorptive masks. This streamlined approach enhances the flexibility and scalability of the SES system, making it more adaptable to various memory storage and retrieval tasks.

In the main paper, we demonstrated a process which permits us to realize 4096 memories. Indeed our approach provides extreme flexibility.

The Referee asks for the capacity of our architecture. Indeed, the number of optical memories which can be installed on our disordered classifier, is essentially limited by the digital memory storing the masks. However in a certain regime (small M^*) our approach provides the advantage in the form of a Lossy compression as documented by the figure in the Correspondent newly added chapter **Data compression with SES** (see also above).

We also characterized the performance of our optical approach with respect to other optical techniques, in particular with respect to Ridge Regression with Speckles. SES shows an increased Classification efficiency with respect to RRS especially when reduced hardware resources are available (small M^*) :

Finally we introduced a table comparing Some characteristics of SES with respect to other optical competitor techniques such as RRS and Diffractive Deep Neural Networks. The comparison is performed in the Comparison with other optical platforms section and in the relative table (here we report for compactness just the Table, While the full chapter is reported in the reply to referee 2 above)

TABLE I. Comparison Between SES, Deep-SES, RRS[21], DDNN[17] . Second column reports about the Write capability(W) and Read capability (R). The third column report efficiency is (defined as the ratio of the number of elements appearing in the diagonal in the confusion matrix with respect to all the elements of the same matrix). Note that for DDNN (efficiency appearing with an asterisk in the third column), the dataset is different that used in SES, Deep-SES, and RRS. "Analog output" column report the possibility to gather the output in the form of intensity. The third column report the possibility of the platform to work as Content Addressable Memory, and to employ Intensity as a measure of the similarity between the stored memory and the probe pattern.

Architecture	Memory Storage	Efficiency ($M^* = 120$)	Analog output	CAM and Similarity
SES	W/R	-	✓	✓
Deep SES	W/R	91.71 %	✓	✓
RRS[21]	none	53.54 %	✗	✗
DDNN[17]	W	91.75* %	✓	✗

- The authors discuss the results presented in this work in relation to the formation processes of biological memory. However, as the authors mentioned, the challenge of understanding the biological memory formation process is a matter of ongoing debate. It is suggested to remove claims related to this aspect as they fall beyond the scope of this work.

We appreciate the referee's perspective and would like to respectfully clarify our stance on the reference to biological memory within our work.

Firstly, our intention in mentioning biological memory was not to make a definitive "claim" per se but rather to establish a conceptual bridge between the SES process and processes related to memory formation in the brain. This connection arises from the intrinsic randomness inherent in SES, which shares similarities with the randomness observed in the strengths of neuronal connections in certain areas of the brain, or at an early stage of the brain growth.

Secondly, we acknowledge the ongoing debate within the scientific community regarding the nature of memory formation, and our aim was to contribute to this discourse by offering a new perspective facilitated by SES. We believe that this perspective can potentially influence and enrich the ongoing debate on this topic.

Lastly, we wish to highlight that our viewpoint on this matter is shared by referee #1, who specifically requested an enhancement of the discussion regarding the "biological memory" aspects.

In the current version of the paper, we have taken the suggestions of referee #1 into account and have indeed augmented the discussion section to further underscore the connection between SES and biological memory formation (see above, referee #1 reply). This revised section aims to provide a more comprehensive exploration of this link, aligning with our goal of contributing to the broader scientific conversation surrounding memory processes.

Also, the combined use of the terms "deterministic" and "random" to describe a phenomenon seems contradictory because determinism and randomness are typically seen as opposing concepts.

We thank the referee for highlighting this point. We modified the text as follows

However, SES goes beyond by exploring the potential of a stochastic *innate* network in which, **pre-existent** random synaptic structures are combined to generate

Minor comments:

- The terms "stochastic Emergent Storage" should be "Stochastic Emergent Storage".
- The terms "stochastic Hebb's storage" should be "Stochastic Hebb's Storage".
- The terms "optical synaptic matrix" should be used consistently throughout the manuscript.
- The acronyms "CAM" and "SES" should be used consistently throughout the manuscript.

- Page 4: the definition of the storage capacity of the proposed learning strategy should be clearly stated.
- Page 4, Figure 3C: the manuscript could be improved by providing a discussion and related references about strategies to improve the signal-to-noise ratio in the classification process of the proposed learning strategy.
- Page 5, the term "Figure 4f" should be "Figure 4e".
- Page 7, Supporting Information, the terms "modulation intensity" should be "modulation, intensity".

We express our gratitude to the referee for pointing out the highlighted typos and errors. We have taken his/her feedback into consideration and made the following improvements:

- To differentiate the constructed optical operator from the naturally occurring "optical synaptic matrix," we have introduced the term "Artificial optical synaptic matrix."
- The error concerning "storage capacity" has been corrected to "storage capability."
- We have included a consistent reference to error correction ([31]) and updated the related text accordingly.
- All figure labeling and typos have been rectified.

These revisions have been implemented to enhance the clarity and accuracy of our work, and we thank the referee for their valuable input.

**** See Nature Portfolio's author and referees' website at www.nature.com/authors for information about policies, services and author benefits.**

This email has been sent through the Springer Nature Tracking System NY-610A-NPG&MTS

Confidentiality Statement:

This e-mail is confidential and subject to copyright. Any unauthorised use or disclosure of its contents is prohibited. If you have received this email in error please notify our Manuscript Tracking System Helpdesk team at <http://platformsupport.nature.com> .

Details of the confidentiality and pre-publicity policy may be found here <http://www.nature.com/authors/policies/confidentiality.html>

Privacy Policy | Update Profile

DISCLAIMER: This e-mail is confidential and should not be used by anyone who is not the original intended recipient. If you have received this e-mail in error please inform the sender and delete it from your mailbox or any other storage mechanism. Springer Nature Limited does not accept liability for any statements made which are clearly the sender's own and not expressly made on behalf of Springer Nature Ltd or one of their agents.

Please note that Springer Nature Limited and their agents and affiliates do not accept any responsibility for viruses or malware that may be contained in this e-mail or its attachments and it is your responsibility to scan the e-mail and attachments (if any).
Springer Nature Ltd. Registered office: The Campus, 4 Crinan Street, London, N1 9XW. Registered Number: 00785998 England.

Reviewer #1 (Remarks to the Author):

I thank the authors for taking their times to going through the concern raised by the reviewers. I believe they have address and improved the content of the paper and thus I do not have further comments.

Reviewer #2 (Remarks to the Author):

I appreciate the efforts made by the authors to address all my comments and provide such a detailed discussion in the response letter. The text flows much better, and the new figures, discussion, and supplementary information (including the benchmarking) make it a compelling article with excellent results. I recommend the current version be published as is.

Reviewer #3 (Remarks to the Author):

In the revised manuscript, the authors have provided a discussion on the implementation and performance of photonic stochastic emergent storage to address the feedback provided by this reviewer.

In the "Full-Hardware SES and Scalability" section, references about the conventional technologies that may be used for full-hardware SES as well as the suggested implementation need to be included.

It is advisable to standardize the formatting of numbers, units of measurement, and punctuation throughout the revisions for enhanced readability.

Issues related to other major and minor comments have been addressed.

REVIEWERS' COMMENTS

Reviewer #1 (Remarks to the Author):

I thank the authors for taking their times to going through the concern raised by the reviewers. I believe they have address and improved the content of the paper and thus I do not have further comments.

We express our gratitude to the referee for their insightful feedback and appreciate their favorable evaluation.

Reviewer #2 (Remarks to the Author):

I appreciate the efforts made by the authors to address all my comments and provide such a detailed discussion in the response letter. The text flows much better, and the new figures, discussion, and supplementary information (including the benchmarking) make it a compelling article with excellent results. I recommend the current version be published as is.

We are thankful to the referee for their valuable insights and positive assessment of our work.

Reviewer #3 (Remarks to the Author):

In the revised manuscript, the authors have provided a discussion on the implementation and performance of photonic stochastic emergent storage to address the feedback provided by this reviewer.

We extend our gratitude to the referee for their contributions, as we believe their feedback significantly aided in enhancing the paper.

In the "Full-Hardware SES and Scalability" section, references about the conventional technologies that may be used for full-hardware SES as well as the suggested implementation need to be included.

Following the referees suggestion we provide an hardware selection (supplementary materials, "Full-Hardware SES and Scalability" section) which enables to realized the full hardware SES

tical complexity. A sketch of the experimental scheme is provided in the Fig. 2 below. As an example an hardware configuration with the performances fulfilling the requirements is: *i)* the DLP9000X texas instument DMD and related DLPC910 controller for both probe and selection DMS; *ii)* FDS100 from thorlabs as large area photodyode, and *iii)* Basler a2A2448-75umPRO camera for mode characterization .

This information, plus figure 2 of the supplementary material providing the hardware SES sketch and the experimental setup details provided in the paper, methods and supplementary information, provide all the information required to realize the full hardware SES.

It is advisable to standardize the formatting of numbers, units of measurement, and punctuation throughout the revisions for enhanced readability.

Upon considering the recommendations from both the referee and editor, we meticulously reviewed the manuscript and enhanced the consistency of text, mathematical symbols, and punctuation across the entire document.

Issues related to other major and minor comments have been addressed.

Once more, we extend our appreciation to the referee for their invaluable contribution to enhancing the manuscript.